# Mamba Unchained: A Permutation-Invariant Approach to Multivariate Time Series

## Abstract

Time series data in domains such as climate science, finance, and biomedicine present a significant challenge for scalable modeling due to their multi-scale temporal patterns, complex inter-variable dependencies, and frequency-specific structures. While recent advances in architectures like Transformers and state space models (SSMs) have shown promise, they are often limited by either high computational costs or an inability to capture time-varying cross-variable interactions. To overcome these limitations, we propose a variable-invariant two-dimensional state space model that eliminates variable ordering dependence by leveraging a global, permutation-invariant descriptor to condition temporal dynamics. This design allows for the efficient processing of variable-axis updates through an effective pooling operation, which maintains global correlations and enables full parallelism. We further enhance this architecture with a multi-branch design that incorporates distinct pathways for long- and short-horizon temporal features and a dedicated frequency-domain pathway, all integrated via a lightweight gating mechanism. Through extensive experiments, our model consistently outperforms state-of-the-art baselines on forecasting, classification, and anomaly detection tasks. Comprehensive analysis confirms our model's efficiency, robustness, and ability to capture diverse temporal–spectral patterns.

## 1 Introduction

Time series data are pervasive in diverse application domains—including climate science (Mudelsee, 2010), finance (Shi, 2024), and biomedical signals (Jeong et al., 2024)—and are characterized by complex temporal dynamics, seasonal regularities, and inter-variable dependencies. In the multivariate setting, such data typically exhibit three salient properties: (i) the coexistence of long-term trends and short-term fluctuations, (ii) intricate and evolving correlations among variables, and (iii) distinct manifestations of these structures in the frequency domain (Qiu et al., 2025). Designing models that simultaneously capture multi-scale temporal patterns, cross-variable relationships, and spectral regularities—while remaining computationally tractable over long horizons—remains an open challenge across tasks such as forecasting (Zhou et al., 2021), classification (Jeong et al., 2023), and anomaly detection (Luo & Wang, 2024).

A broad spectrum of neural architectures has been explored to address this challenge. CNN- and RNN-based models achieved early success by capturing local patterns and autoregressive dependencies, but they were less effective for long-range modeling (Shen et al., 2020; Franceschi et al., 2019). Transformer-based architectures introduced powerful global context modeling through self-attention and domain-specific inductive biases (Wen et al., 2023), yet the quadratic complexity of attention with sequence length imposes severe computational overhead, limiting their scalability in practice (Wang et al., 2025).

Recently, state space models (SSMs) have emerged as compelling alternatives, offering structured recurrences that allow linear-time inference while achieving competitive accuracy (Gu et al., 2021). Among them, Mamba (Gu & Dao, 2023)—an input-selective SSM—has demonstrated strong performance across a variety of time-series benchmarks (Zeng et al., 2024). However, conventional one-dimensional SSMs evolve only along the temporal axis, limiting their ability to account for dynamic inter-variable relations. Extensions such as CMamba (Zeng et al., 2024) and MambaTS (Cai et al., 2024) partially mitigate this issue by incorporating channel-level structure or task-specific

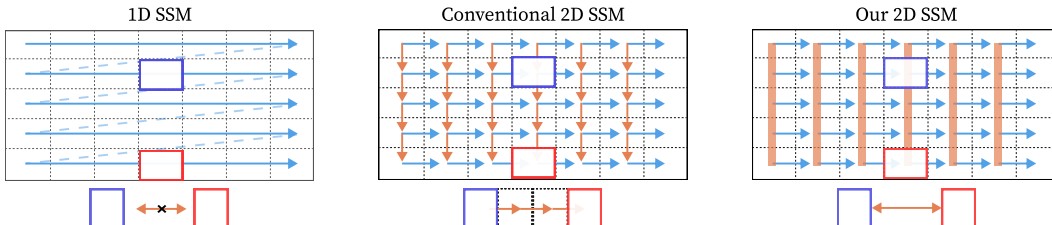

Figure 1: **Left:** 1D SSM models only along the temporal axis, overlooking dependencies across variables. **Middle:** Conventional 2D SSM captures inter-variable correlations through sequential scans, but still imposes artificial ordering and struggles with distant relationships. **Right:** Our method performs global aggregation over variables, enabling simultaneous and order-free modeling of inter-variable relationships over time.

priors, but they still fall short of modeling time-varying dependencies across variables in a generalizable way.

This limitation has motivated two-dimensional extensions of SSMs, initially introduced in vision tasks to alleviate spatial mismatches (Zhang et al., 2025; Baron et al., 2024) and more recently adapted to multivariate time series (Behrouz et al., 2024). By coupling updates along both temporal and variable axes, such models enable richer representations of evolving variable interactions. Yet, these methods retain an inherent drawback: due to the causal nature of SSMs, they rely on sequential scanning along the variable axis, which induces artificial ordering dependencies (Figure 1). While ordering is natural in spatial domains such as images, multivariate time series lack a canonical variable arrangement. As a result, sequential scanning not only imposes unnecessary inductive bias but also reduces robustness to permutation and hinders parallel computation when the number of variables is large.

To overcome these challenges, we propose a variable-invariant two-dimensional state space model. Our formulation replaces sequential updates along the variable axis with a global, permutation-invariant aggregation of variable states that conditions the temporal dynamics. This design eliminates ordering dependence, preserves global inter-variable relations, and reduces the 2D pass to a temporal scan with a single parallel pooling operation over variables, thereby enabling full parallelism. Building upon this foundation, we introduce a multi-branch architecture comprising two temporal pathways (long- and short-horizon) and a frequency-domain pathway. A lightweight gating mechanism adaptively fuses these pathways, enhancing expressivity across heterogeneous temporal and spectral patterns. Extensive experiments on forecasting, classification, and anomaly detection demonstrate that our approach consistently outperforms state-of-the-art baselines. Beyond accuracy, the proposed method delivers substantial gains in computational efficiency and robustness, particularly in settings with many variables or under arbitrary variable orderings. These results highlight the promise of permutation-invariant 2D SSMs as a scalable and general framework for modeling complex multivariate time series.

Our contributions are summarized as follows:

**Variable-invariant 2D SSM.** We introduce a new 2D SSM formulation that eliminates forced ordering along the variable axis via a global pooling operator. This enables permutation-invariant modeling of variable interactions, reduces the 2D scan to a 1D temporal scan with a single parallel pooling step across variables, and improves both computational efficiency and scalability.

**Multi-branch 2D Mamba architecture.** Building on this formulation, we propose a novel Mamba variant tailored for multivariate time series. The architecture incorporates three complementary pathways—long-term, short-term, and frequency—each instantiated with 2D SSMs, and fuses them through input-adaptive gating. This design enhances expressivity for heterogeneous temporal and spectral dynamics.

**Comprehensive empirical validation.** We evaluate the proposed method on forecasting, classification, and anomaly detection benchmarks, where it achieves superior performance over state-of-the-art baselines. Through detailed ablations and case studies, we confirm the contribution of each architectural component and demonstrate the advantages of permutation invariance in both accuracy and efficiency.

## 2 RELATED WORKS

Research on deep learning for multivariate time series has evolved from modeling purely temporal dynamics to explicitly capturing cross-variable dependencies (Qiu et al., 2025). Early CNN- and RNN-based methods extracted local patterns and autoregressive structures but were limited in modeling long-range dependencies (Lai et al., 2018; Shen et al., 2020; Franceschi et al., 2019; He & Zhao, 2019). Transformer-based architectures advanced global context modeling through self-attention, yet their quadratic complexity with sequence length remains a scalability bottleneck, even with sparsification and kernelization attempts (Keles et al., 2023). To improve efficiency, convolutional alternatives (Luo & Wang, 2024) and SSMs (Gu et al., 2021; Gu & Dao, 2023; Patro & Agneeswaran, 2024) have been proposed, offering linear or near-linear complexity. Still, most existing formulations evolve solely along the temporal axis and struggle to capture time-varying cross-variable interactions.

To overcome this limitation, extensions of Mamba such as TimePro (Ma et al.), CMamba (Zeng et al., 2024), Simba (Patro & Agneeswaran, 2024), ET-Mamba (Jeong et al., 2025), and GrootV-T (Xiao et al., 2024) incorporate variable-level priors but often rely on heuristics. This has motivated two-dimensional SSMs, inspired by classical formulations such as Roesser's model (Kung et al., 1977). Recent variants (Baron et al., 2024; Zhang et al., 2025; Behrouz et al., 2024) couple temporal and variable dynamics to better capture multivariate correlations. Among them, Chimera (Behrouz et al., 2024) enhances cross-variable modeling but still requires sequential scans along the variable axis, introducing ordering dependence and limiting parallelism. These limitations motivate our variable-invariant 2D SSM.

## 3 PRELIMINARIES

**Problem Definition.** Let $X \in \mathbb{R}^{C \times T}$ denote a multivariate time series with $C$ variates and temporal length $T$. We index time by $t \in \{1, \ldots, T\}$ and variables by $c \in \{1, \ldots, C\}$, where the observation at position $(t, c)$ is denoted by $x(t, c)$. The goal is to learn a mapping $f$ that extracts informative representations from a context window $X_{1:T}$ to support a broad range of downstream tasks, including forecasting, classification, and anomaly detection. Motivated by two-dimensional dynamical structure, we consider a 2D state-space architecture with two coupled latent states: a *horizontal* state $h_h(t, c)$ evolving along the temporal axis and a *vertical* state $h_v(t, c)$ evolving along the variable axis. Our objective is to preserve rich inter-variable dependencies while avoiding artificial ordering assumptions on the variable axis (*i.e.*, to achieve *permutation invariance*).

**Roesser's 2D State Space Models.** The classical extension of 1D SSMs to multidimensional systems includes the Roesser model (Kung et al., 1977) and its variants (Eising, 2003; Fornasini & Marchesini, 1978; Givone & Roesser, 2006). In the Roesser formulation, widely used in recent deep 2D architectures (Baron et al., 2024; Behrouz et al., 2024; Zhang et al., 2025), horizontal/vertical dynamics are coupled as:

$$\frac{\partial h_h(t, c)}{\partial t} = (A_h h_h(t, c), \ A_{hv} h_v(t, c)) + B_h x(t, c), \tag{1}$$

$$\frac{\partial h_v(t, c)}{\partial c} = (A_v h_v(t, c), \ A_{vh} h_h(t, c)) + B_v x(t, c), \tag{2}$$

$$y(t, c) = C_h h_h(t, c) + C_v h_v(t, c). \tag{3}$$

Here, $h_h$ and $h_v$ represent the horizontal (time) and vertical (variable) states, respectively, which evolve jointly through coupled recurrences. This formulation provides a principled means of modeling temporal evolution alongside variable-wise interactions and serves as the foundation for modern 2D SSM-based neural architectures.

**2D Mamba for Multivariate Time Series.** Recent advances in selective-scan SSMs, such as Mamba, have motivated 2D generalizations tailored to long-range temporal dependencies and cross-variable structure (Behrouz et al., 2024; Zhang et al., 2025). Chimera (Behrouz et al., 2024) exem-

plifies this direction by extending selective-scan dynamics to both time and variable axes:

$$h_h[t+1, c] = \bar{A}_1 h_h[t, c] + \bar{B}_1 x[t+1, c], \tag{4}$$

$$h_v[t, c+1] = \bar{A}_2 h_v[t, c] + \bar{B}_2 x[t, c+1], \tag{5}$$

$$y_{v,t} = C_1 h_h + C_2 h_v, \tag{6}$$

with zero-order-hold (ZOH) discretization, $\bar{A}_i = \exp(\Delta_i A_i)$ and $\bar{B}_i = A_i^{-1}(\bar{A}_i - I)B_i, \forall i \in \{1, 2\}$. This design enables simultaneous modeling along time and variable axes using selective scan kernels. However, whereas time has an inherent causal direction, most datasets lack a canonical causal order across variables. Consequently, *variable-axis scanning induces an artificial ordering constraint and prevents the model from aggregating global variable information at each time step*. This limits generality and parallel efficiency, especially when performance hinges on capturing exchangeable, global inter-variable statistics.

These challenges motivate the variable-invariant 2D SSM proposed in the next section: it preserves horizontal–vertical coupling via a global, permutation-invariant summary while eliminating the need for sequential scanning along the variable axis.

## 4 PROPOSED METHODS

Permutation-invariant modeling has a long history in set learning—*e.g.*, Deep Sets (Zaheer et al., 2017) and Set Transformer (Lee et al., 2019) enforce invariance/equivariance via pooling or attention over unordered elements. Our work brings this principle to 2D SSMs: rather than scanning over variables, we replace the vertical derivative with a global, permutation-invariant summary that conditions the temporal SSM. This hybridization preserves global coupling between horizontal and vertical states while collapsing the 2D pass to a 1D selective scan plus $O(C)$ pooling (a parallel aggregation across all variables). These crucial characteristics are comparable to prior 2D SSMs (Behrouz et al., 2024), the resulting dynamics are provably permutation-equivariant on the variable axis, unlock full parallelism across variables, and maintain the data-dependent parameterization.

### 4.1 VARIABLE-INVARIANT 2D SSM

Let $h_v(t, :) = [h_v(t, 1), \ldots, h_v(t, C)] \in \mathbb{R}^{C \times d_v}$ be the vertical states at time $t$. To eliminate variable ordering dependencies, we replace the vertical derivative with a global information

$$\psi(t) := \phi(\{W_v h_v(t, c)\}_{c=1}^C), \tag{7}$$

where $W_v$ is a learnable weight and $\phi$ is a permutation-invariant set aggregator (*e.g.*, mean/sum pooling, gated pooling, or set attention that aggregates by a symmetric operation over elements).

**Proposition 4.1** (Permutation invariance of the pooled summary). *Let $\pi$ be any permutation of $\{1, \ldots, C\}$ and define $\psi_\pi(t) := \phi(\{W_v h_v(t, \pi(c))\}_{c=1}^C)$. Assume (i) $W_v$ is variable-shared (independent of c), and (ii) $\phi$ is permutation-invariant on multisets, i.e., $\phi(\{z_c\}_{c=1}^C) = \phi(\{z_{\pi(c)}\}_{c=1}^C)$ for any collection $\{z_c\}$. Then $\psi_\pi(t) = \psi(t)$ for all t, where $\psi(t) = \phi(\{W_v h_v(t, c)\}_{c=1}^C)$.*

By the commutative law of $\phi$, for any variable permutation $\psi_\pi = \psi$, thus the entire model has a representation that is invariant to variable order. This configuration allows for simultaneous updates along the variable axis, enabling fully parallel computation across variables rather than sequential scanning. See proof in Appendix C.1.

**Modified Dynamics with Global Coupling.** We augment the 2D-SSM in Sec. 3 by injecting a global, permutation-invariant summary $\psi(t)$ into both horizontal and vertical dynamics:

$$\frac{\partial h_h(t, c)}{\partial t} = A_h h_h(t, c) + A_{h\psi}\psi(t) + B_h x(t, c), \tag{8}$$

$$\frac{\partial h_v(t, c)}{\partial t} = A_v h_v(t, c) + A_{v\psi}\psi(t) + A_{vh}h_h(t, c) + B_v x(t, c). \tag{9}$$

Here, $A_{h\psi}$ and $A_{v\psi}$ couple the two sub-states through the global summary $\psi(t)$. In contrast to Roesser's model, the variable-axis derivative $\partial/\partial c$ is removed, so no artificial ordering over variables is required; global inter-variable interactions are instead mediated via $\psi(t)$. In line with stability/expressivity considerations in prior SSM work, we further constrain the transition structure (*e.g.*, companion) for $A_i$ to capture long-range temporal dependencies while keeping discretization well-conditioned effectively.

**Proposition 4.2.** *Assume $A_h, A_v, A_{h\psi}, A_{v\psi}, A_{vh}, B_h, B_v$ are variable-shared (independent of c), and let $\psi(t)$ be defined as above with a permutation-invariant $\phi$ (Prop. 4.1). For any permutation $\pi$ of $\{1, \ldots, C\}$, consider the permuted inputs $x^\pi(t, c) := x(t, \pi(c))$ and permuted initial states $h^\pi(0, c) := h(0, \pi(c))$. Then the unique solutions to Eq. 8-9 satisfy $h_h^\pi(t, c) = h_h(t, \pi(c))$, $h_v^\pi(t, c) = h_v(t, \pi(c)) \ \forall \ t, c$, i.e., the dynamics are permutation equivariant with respect to the variable axis.*

With variable-shared parameters, the permuted system is a mere relabeling of indices; hence, the solutions are correspondingly permuted. See proof in Appendix C.2.

**Discrete 2D SSM.** We discretize the continuous, variable-invariant dynamics in Eq. 8-9 with a ZOH. Let the sampling step be $\Delta > 0$, and assume the exogenous input $x(t, c)$ and the global information $\psi(t)$ are held constant on each interval $[k\Delta, (k+1)\Delta)$:

$$x(t, c) = x[k, c], \quad \psi(t) = \psi[k] \quad \text{for } t \in [k\Delta, (k+1)\Delta). \tag{10}$$

We also define the discrete global information by the same permutation-invariant pooling. Then the ZOH-discretized updates are

$$h_h[t+1, c] = \bar{A}_h h_h[t, c] + \bar{B}_h^\psi \psi[t] + \bar{B}_h^x x[t+1, c], \tag{11}$$

$$h_v[t+1, c] = \bar{A}_v h_v[t, c] + \bar{A}_{vh} h_h[t, c] + \bar{B}_v^\psi \psi[t] + \bar{B}_v^x x[t+1, c], \tag{12}$$

$$y = C_h h_h + C_v h_v, \tag{13}$$

where $\bar{A}_i = \exp(\Delta A_i)$, $\bar{B}_i^\psi = A_i^{-1}(\bar{A}_i - I)A_{i\psi}$, $\bar{B}_i^x = A_i^{-1}(\bar{A}_i - I)B_i$, $i \in \{h, v\}$. The closed-loop dependence through $\psi[k]$ is algebraic (no extra recurrence in $c$), hence the update over variables remains parallel and permutation-invariant. Detailed derivation and convolution formulation can be found in Appendix B.

**Data-Dependent Parameters.** Recent SSM advances (*e.g.*, Mamba) emphasize input-conditioned dynamics, where transition/measurement vectors and step sizes adapt to the current input. Furthermore, models need to adaptively learn the importance of temporal variations based on the data. We follow this principle and make the discrete SSM parameters data dependent:

$$B[t, c] = f_\theta^B(x[t, c]), \quad C[t, c] = f_\theta^C(x[t, c]), \quad \Delta[t] = f_\theta^\delta(x[t, :]). \tag{14}$$

Here, $f_\theta$ is a pointwise map applied per variable and time; their outputs have the same shape as the corresponding SSM vectors.

**2D Selective Scan.** Although our model is a 2D SSM, it exploits the variable-invariant construction to avoid an explicit vertical scan. At each time $t$, we compute the permutation-invariant summary $\psi[t]$ and then perform a hybrid scan:

- a 1D selective scan along the time axis (causal, efficient, input-conditioned via $\Delta[t], B[t, c], C[t, c]$,
- coupled with global pooling on the variable axis to transmit cross-variable information.

This replaces the inefficient $O(T \times C)$ variable-axis sequential pass with an $O(T)$ time-only scan that is permutation-invariant in $c$, while still preserving global inter-variable coupling through $\psi[t]$. In practice, $\psi[t]$ can be implemented as mean/sum pooling, gated pooling, or attention pooling; the choice leaves the invariance intact and controls the strength and bandwidth of variable-wise interactions.

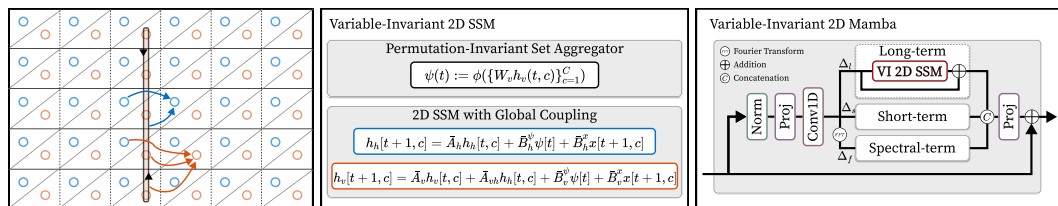

Figure 2: Overview of the proposed Variable-Invariant 2D Mamba. A permutation-invariant global descriptor $\psi(t)$ replaces variable-axis recurrence, enabling a parallel 2D SSM block that fuses long-term, short-term, and spectral pathways.

## 4.2 VARIABLE-INVARIANT 2D MAMBA

In Section 4.1, we introduced the formulation of our 2D SSM, which provides a foundation for modeling inter-variable relationships that evolve over time. Building on this, we now present the neural architecture of our variable-invariant 2D Mamba, designed to capture both multi-scale temporal dynamics and spectral information that are characteristic of diverse time-series domains. Specifically, our architecture integrates: (i) long- and short-term modeling in the time domain, and (ii) frequency-level representations derived from spectral analysis.

### 4.2.1 LONG- AND SHORT-TERM PATTERNS IN THE TIME DOMAIN

Time-series data inherently exhibit structure at multiple scales. Depending on the task and domain, informative signals may range from local, fine-grained dependencies between adjacent time points to long-term, global trends spanning extended horizons. To capture such patterns, recent research has emphasized multi-scale temporal modeling as a central design principle (Behrouz et al., 2024; Karadag et al., 2025).

In our framework, multi-scale dynamics are realized by adjusting the discretization parameter $\Delta$ during SSM discretization. Following prior work (Behrouz et al., 2024; Karadag et al., 2025), $\Delta$ can be interpreted as the effective time resolution or sampling rate in continuous-time formulations. We therefore instantiate two temporal pathways: a long-term path parameterized by $\Delta_l$ and a short-term path parameterized by $\Delta_s$, with $\Delta_l \gg \Delta_s$. Larger $\Delta$ values correspond to slower state updates, effectively capturing long-range trends, whereas smaller $\Delta$ values yield faster dynamics suited for modeling rapid fluctuations. The long-term path captures broader temporal dependencies by operating at a coarser resolution, whereas the short-term path preserves fine-grained temporal detail. Together, these pathways yield a complementary multi-scale representation of temporal dynamics.

### 4.2.2 SPECTRAL INFORMATION IN THE FREQUENCY DOMAIN

For many time-series modalities, the frequency domain plays a central role in characterizing the underlying signal (Qiu et al., 2025). Frequency-domain features complement time-domain representations: low-frequency components often capture global, slowly varying dynamics, whereas high-frequency components reveal localized, transient variations. To leverage this complementary structure, we introduce a frequency-domain pathway into our architecture.

Concretely, the input sequence is transformed into the frequency domain via the Fourier Transform. After appropriate scaling, the transformed representation is processed by the proposed 2D SSM, which in this case operates across frequency rather than time as the scan dimension.

**Interpretation of SSM in the Frequency Domain.** In this setting, the semantics of the SSM differ from the temporal case. Rather than modeling evolution over time, the states evolve across frequency bands, capturing dependencies between variables across spectral ranges. Importantly, the update operates as a continuous-time SSM applied to the frequency axis, where the discretization step size $\Delta_f$ governs how densely the underlying spectral ODE is sampled. Unlike time, frequency does not exhibit causality; it spans from low to high values, with most signal energy concentrated at low frequencies and increasingly sparse but informative oscillatory patterns at higher frequencies.

This spectral imbalance imposes a numerical challenge. If $\Delta_f$ is large, the discretization becomes too coarse; the exponential term in the SSM update inaccurately scales higher-frequency modes, causing instability, aliasing, and over-attenuation of small but critical spectral components. Setting $\Delta_f$ sufficiently small stabilizes the SSM dynamics, suppresses aliasing, and increases the effective resolution in the high-frequency range, ensuring that transient or oscillatory structure is preserved despite low energy levels. In practice, we find that values of $\Delta_f$ within $[0.001, 0.01]$ offer a robust trade-off between spectral fidelity and numerical stability.

### 4.2.3 MULTI-DOMAIN FEATURE AGGREGATION

Our final architecture integrates information from three complementary pathways: the long-term temporal branch ($\Delta_l$), the short-term temporal branch ($\Delta_s$), and the frequency-domain branch ($\Delta_f$). The outputs of these branches are fused by a learnable gating vector that adaptively weights each branch before aggregation, enabling dynamic emphasis depending on the input characteristics. This tripartite design ensures complementary coverage of temporal scales and spectral content, improving generalization across heterogeneous time-series tasks, as summarized in Algorithm 2.

## 5 EXPERIMENTS

To evaluate the validity and generality of the proposed method, we conducted experiments across four representative tasks: long-term forecasting, short-term forecasting, classification, and anomaly detection. The proposed method was extensively compared against state-of-the-art baselines, including Transformer-based approaches (Liu et al., 2023; Nie et al., 2022; Zhang & Yan, 2023) and Mamba-based method (Patro & Agneeswaran, 2024; Behrouz et al., 2024). Detailed experimental settings and additional results are provided in the Appendix F and Github[1].

Table 1: Results of long-term time-series forecasting. Best results are highlighted in **bold**, and second-best results are underlined.

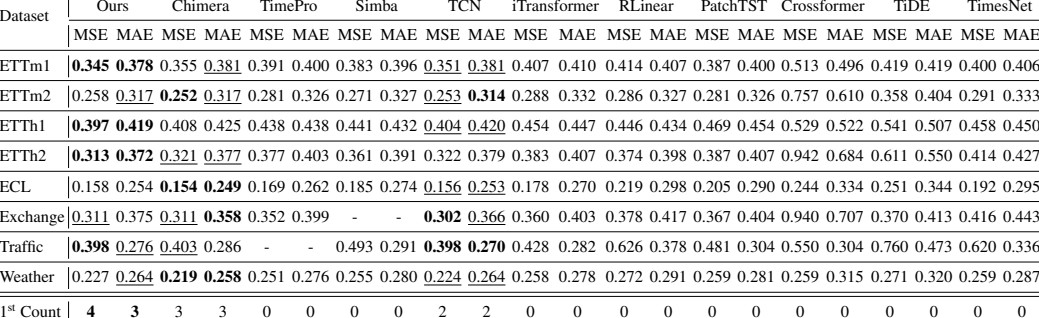

| Dataset | Ours | | Chimera | | TimePro | | Simba | | TCN | | iTransformer | | RLinear | | PatchTST | | Crossformer | | TiDE | | TimesNet | |
|---|---|---|---|---|---|---|---|---|---|---|---|---|---|---|---|---|---|---|---|---|---|---|
| | MSE | MAE | MSE | MAE | MSE | MAE | MSE | MAE | MSE | MAE | MSE | MAE | MSE | MAE | MSE | MAE | MSE | MAE | MSE | MAE | MSE | MAE |
| ETTm1 | **0.345** | **0.378** | 0.355 | 0.381 | 0.391 | 0.400 | 0.383 | 0.396 | 0.351 | 0.381 | 0.407 | 0.410 | 0.414 | 0.407 | 0.387 | 0.400 | 0.513 | 0.496 | 0.419 | 0.419 | 0.400 | 0.406 |
| ETTm2 | 0.258 | 0.317 | **0.252** | 0.317 | 0.281 | 0.326 | 0.271 | 0.327 | 0.253 | **0.314** | 0.288 | 0.332 | 0.286 | 0.327 | 0.281 | 0.326 | 0.757 | 0.610 | 0.358 | 0.404 | 0.291 | 0.333 |
| ETTh1 | **0.397** | **0.419** | 0.408 | 0.425 | 0.438 | 0.438 | 0.441 | 0.432 | 0.404 | 0.420 | 0.454 | 0.447 | 0.446 | 0.434 | 0.469 | 0.454 | 0.529 | 0.522 | 0.541 | 0.507 | 0.458 | 0.450 |
| ETTh2 | **0.313** | **0.372** | 0.321 | 0.377 | 0.377 | 0.403 | 0.361 | 0.391 | 0.322 | 0.379 | 0.383 | 0.407 | 0.374 | 0.398 | 0.387 | 0.407 | 0.942 | 0.684 | 0.611 | 0.550 | 0.414 | 0.427 |
| ECL | 0.158 | 0.254 | **0.154** | **0.249** | 0.169 | 0.262 | 0.185 | 0.274 | 0.156 | 0.253 | 0.178 | 0.270 | 0.219 | 0.298 | 0.205 | 0.290 | 0.244 | 0.334 | 0.251 | 0.344 | 0.192 | 0.295 |
| Exchange | 0.311 | 0.375 | 0.311 | **0.358** | 0.352 | 0.399 | - | - | **0.302** | 0.366 | 0.360 | 0.403 | 0.378 | 0.417 | 0.367 | 0.404 | 0.940 | 0.707 | 0.370 | 0.413 | 0.416 | 0.443 |
| Traffic | **0.398** | 0.276 | 0.403 | 0.286 | - | - | 0.493 | 0.291 | **0.398** | **0.270** | 0.428 | 0.282 | 0.626 | 0.378 | 0.481 | 0.304 | 0.550 | 0.304 | 0.760 | 0.473 | 0.620 | 0.336 |
| Weather | 0.227 | 0.264 | **0.219** | **0.258** | 0.251 | 0.276 | 0.255 | 0.280 | 0.224 | 0.264 | 0.258 | 0.278 | 0.272 | 0.291 | 0.259 | 0.281 | 0.259 | 0.315 | 0.271 | 0.320 | 0.259 | 0.287 |
| 1st Count | **4** | **3** | 3 | 3 | 0 | 0 | 0 | 0 | 2 | 2 | 0 | 0 | 0 | 0 | 0 | 0 | 0 | 0 | 0 | 0 | 0 | 0 |

### 5.1 LONG-TERM FORECASTING

**Settings.** We conducted experiments on eight benchmark datasets commonly used in time-series data forecasting tasks, including the Weather, Traffic, Electricity, Exchange, and 4 ETT datasets (Zhou et al., 2021). Baseline results were obtained from the corresponding literature to ensure consistency of comparison, and all models were evaluated using mean squared error (MSE) and mean absolute error (MAE). For brevity, we report the average results in the main text, while detailed results and extended baseline comparisons are provided in Table 10.

**Results.** Table 1 highlights the effectiveness of the proposed method. It achieves the lowest MSE on four of the eight datasets and the lowest MAE on three, yielding the best overall performance among all baselines. Compared to Transformer-based methods, our approach consistently outperforms across all datasets, and it also surpasses 1D SSM variants (Patro & Agneeswaran, 2024; Ma et al.). Against Chimera (Behrouz et al., 2024), a 2D SSM model, the proposed method achieves

---

[1]https://anonymous.4open.science/r/Variable-Invariant-2D-SSM-1346

superior results on most datasets and remains competitive on the others, further demonstrating its robustness and effectiveness.

Table 2: Results of short-term time-series forecasting in the M4 dataset. Best results are highlighted in **bold**, and second-best results are underlined.

| Weighted Avg. | Ours | Chimera | ModernTCN | PatchTST | TimesNet | N-HiTS | N-BEATS | ETSformer | LightTS | DLinear | FEDformer |
|---|---|---|---|---|---|---|---|---|---|---|---|
| SMAPE | 11.686 | **11.618** | 11.698 | 11.807 | 11.829 | 11.927 | 11.851 | 14.718 | 13.525 | 13.639 | 12.840 |
| MASE | 1.549 | **1.528** | 1.556 | 1.590 | 1.585 | 1.613 | 1.599 | 2.408 | 2.111 | 2.095 | 1.701 |
| OWA | 0.832 | **0.827** | 0.838 | 0.851 | 0.851 | 0.861 | 0.855 | 1.172 | 1.051 | 1.051 | 0.918 |

## 5.2 SHORT-TERM FORECASTING

**Settings.** Short-term forecasting was evaluated on the widely adopted M4 dataset (Makridakis et al., 2018). Performance was assessed using three standard metrics—Symmetric Mean Absolute Percentage Error (SMAPE), Mean Absolute Scaled Error (MASE), and Overall Weighted Average (OWA)—with baseline results drawn directly from the original literature to ensure comparability. For clarity, we report only the weighted average results in the main text, while the complete set of results is deferred to Table 11.

**Results.** Table 2 reports the results on the M4 dataset. While the proposed method did not obtain the best score, it achieved the second-best performance, indicating its ability to effectively capture short-term patterns. Its slightly lower performance relative to Chimera is attributable to the dataset's single-channel setting, where the advantages of variable-invariant modeling are less pronounced compared to Chimera's inherent 2D formulation. Nonetheless, our method surpasses other state-of-the-art baselines, confirming its competitiveness in short-term forecasting.

## 5.3 CLASSIFICATION & ANOMALY DETECTION

**Settings.** For classification and anomaly detection, we benchmarked the proposed method on ten datasets from the UEA archive (Bagnall et al., 2018) and five widely adopted anomaly detection datasets: SMD (Su et al., 2019), SWaT (Mathur & Tippenhauer, 2016), PSM (Abdulaal et al., 2021), MSL, and SMAP (Hundman et al., 2018). Classification performance was assessed using overall accuracy, while anomaly detection was evaluated with precision, recall, and F1 score. Figure 3 presents the aggregated results, reporting average accuracy (for classification) and average F1 score (for anomaly detection). Comprehensive results with detailed per-dataset comparisons are provided in Tables 12 and 13.

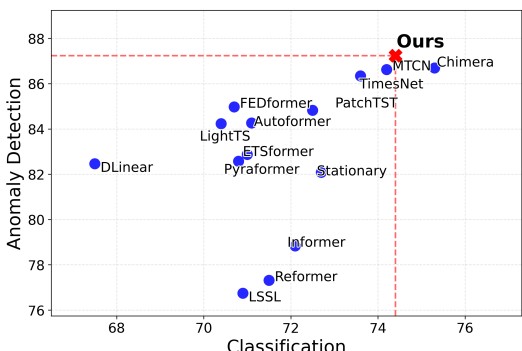

Figure 3: Results of classification (accuracy) and anomaly detection (F1 score).

**Results.** Figure 3 summarizes the overall results. Our proposed method achieves the best performance in anomaly detection, and while it falls slightly short of Chimera in classification, it remains superior to other baselines. This difference can be attributed to the nature of the tasks: anomaly detection strongly benefits from permutation-invariant modeling, as rare deviations often manifest across variable interactions without being tied to a fixed ordering. In contrast, classification tasks in the UEA datasets often involve structured temporal features with relatively limited variable dimensionality, where Chimera's explicit sequential modeling of the variable axis may still provide an advantage. Although classification accuracy is marginally lower than Chimera, our method delivers competitive results at substantially lower computational cost, making it an attractive and efficient alternative—particularly given its clear gains in anomaly detection.

## 6 ANALYSIS

To evaluate the superiority and efficiency of the proposed method, we conducted three complementary analyses. Specifically, we analyzed (i) how efficient the proposed 2D SSM is compared to existing 2D SSMs, (ii) how well our proposed method performs in case studies where inter-variable relationships are important, and (iii) how each module of the proposed Mamba contributes to the overall performance.

**Efficiency of the Proposed 2D SSM.** To assess the computational efficiency of our formulation, we compared it against a 1D SSM and the 2D SSM (Behrouz et al., 2024). For a fair comparison, all models shared identical configurations, differing only in the SSM component. We measured training efficiency using sec/epoch (lower is better) and throughput (higher is better), while varying the number of variables at a fixed sequence length of 256.

As shown in Figure 4, our method achieves efficiency close to that of the 1D SSM, with only a slight overhead attributable to global variable aggregation. By contrast, conventional 2D SSM exhibits a sharp decline in efficiency as the number of variables increases, highlighting the scalability bottleneck introduced by sequential scans. These results confirm that the proposed formulation retains the efficiency properties discussed in Section 4.1 while eliminating the inefficiencies of conventional 2D SSMs.

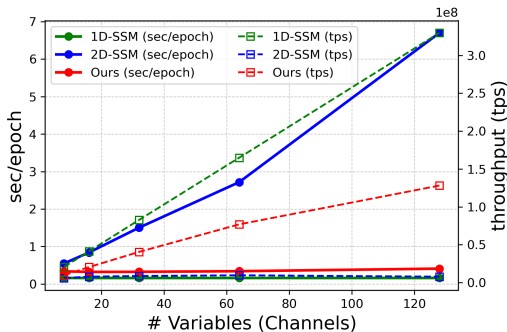

Figure 4: Efficiency analysis with respect to the number of variables.

To clarify computational efficiency, we provide a quantitative comparison of FLOPs and peak GPU memory across representative transformer- and SSM-based baselines using their official implementations. Our model requires only a single temporal SSM scan plus a parallel pooling step, yielding substantially lower FLOPs than methods that recursively evolve states along both axes. A full derivation and empirical results are reported in Appendix E.3.

**Controlled Simulation.** To evaluate accuracy in settings where inter-variable relationships are critical, we constructed a controlled simulation using a VAR(1) process defined on a Watts–Strogatz small-world graph (Watts, 1998). The dataset comprised 64 variables over 1000 time steps, and forecasting performance was assessed using MAE and MAPE (Table 3). To further investigate scalability with respect to the number of variables, we performed a controlled $C$-scaling experiment by varying $C \in \{16, 32, 64, 128, 256\}$ and measuring both performance and computational efficiency (Table 4).

Under the baseline setting (no permutation), the proposed method and a conventional 2D-SSM achieved comparable MAE/MAPE, indicating that both formulations can capture the underlying dynamics when a favorable variable ordering is provided. Nevertheless, our method was approximately 3.8× faster per epoch, reflecting the efficiency gain from removing the variable-axis scan.

Table 3: Controlled simulation for variable-ordering robustness.

| Setting | sec/epoch | MAE (std) | MAPE (std) |
|---|---|---|---|
| Ours | 6.1s | 0.093 (0.000) | 2.007 (0.000) |
| + *Permutation* | | 0.093 (0.000) | 2.063 (0.050) |
| 2D-SSM | 23.4s | 0.093 (0.000) | 1.954 (0.000) |
| + *Permutation* | | 0.094 (0.001) | 2.341 (0.130) |

Table 4: Controlled simulation for $C$-scaling.

| Method | # C | sec/epoch | MAE | MAPE |
|---|---|---|---|---|
| Ours | 16 | 6.2s | 0.106 | 2.465 |
| | 32 | 6.1s | 0.098 | 2.146 |
| | 64 | 6.1s | 0.093 | 2.007 |
| | 128 | 6.4s | 0.096 | 2.313 |
| | 256 | 6.3s | 0.097 | 2.663 |
| 2D-SSM | 16 | 13.4s | 0.105 | 2.603 |
| | 32 | 19.4s | 0.098 | 2.101 |
| | 64 | 23.4s | 0.093 | 1.954 |
| | 128 | 51.0s | 0.096 | 2.489 |
| | 256 | 88.8s | 0.097 | 2.779 |

To assess robustness to variable ordering, we repeated training and evaluation over 10 random permutations of the variable indices. The proposed method exhibited negligible performance drift

across permutations (low variance), consistent with its permutation-invariant design via the global summary $\psi$. In contrast, the conventional 2D-SSM showed significantly larger performance variance, revealing sensitivity to the imposed ordering. This suggests that while 2D-SSM can achieve competitive accuracy under favorable indexings, its performance degrades and variability increases under re-orderings.

Beyond permutation robustness, the proposed formulation also exhibits favorable scaling behavior as the number of variables increases. While conventional 2D-SSMs require a recursive pass along the variable axis, our model replaces this sequence of updates with a single parallel pooling step, whose cost scales linearly in theory but executes as a fully batched GPU operation in practice. Both methods achieved comparable MAE/MAPE at all dimensionalities, confirming that eliminating variable-axis recurrence does not compromise expressiveness. However, while the runtime of standard 2D-SSM grew almost linearly with $C$, our method maintained nearly constant training time, demonstrating that the benefit of variance-invariant pooling intensifies as dimensionality grows. This highlights that the proposed formulation is particularly advantageous in high-dimensional multivariate systems where cross-variable interactions coexist with scalability demands.

**Ablation Study.** In Table 5, we systematically evaluate the contribution of each component of the proposed Mamba model through three ablation cases: (Case I) analysis of individual branches, (Case II) assessment of frequency-domain resolution under varying $\Delta_f$ settings, and (Case III) comparison of permutation-invariant aggregation functions $\psi$.

In Case I, removing any branch led to performance degradation, confirming that all modules contribute meaningfully. Among them, the short-term and long-term pathways have the most significant impact, while the frequency branch plays a complementary role. Case II further highlights this complementarity: although

Table 5: Ablation study on ETTh1 and ETTm1. **Case I**: branch removal; **Case II**: different $\Delta_f$ ranges in the frequency domain.

| Setting | | ETTh1 | | ETTm1 | |
|---|---|---|---|---|---|
| | | MSE | MAE | MSE | MAE |
| I | w/o short | 0.433 | 0.440 | 0.349 | 0.381 |
| | w/o long | 0.431 | 0.439 | 0.350 | 0.381 |
| | w/o freq. | 0.425 | 0.437 | 0.348 | 0.381 |
| II ($\Delta_f$) | $[0.1, 0.5]$ | 0.406 | 0.426 | 0.349 | 0.380 |
| | $[0.01, 0.05]$ | 0.404 | 0.423 | 0.349 | 0.380 |
| III ($\psi$) | Mean | 0.397 | 0.419 | 0.345 | 0.378 |
| | Attention | 0.409 | 0.426 | 0.347 | 0.381 |
| Ours | | 0.397 | 0.419 | 0.345 | 0.378 |

the best results are obtained when all branches are combined, performance gradually improves as $\Delta_f$ decreases, indicating that finer resolution in the high-frequency region enhances modeling capacity. In Case III, mean pooling consistently outperforms attention-based pooling across both datasets. This supports our design choice: forecasting tasks benefit more from stable, low-variance, parameter-free aggregation, with minimal computational overhead. This ablation confirms that the default choice of mean pooling for forecasting tasks is both empirically justified and computationally optimal. A detailed discussion of task-dependent aggregation choices and their computational properties is provided in Appendix E.

# 7 CONCLUSION

In this work, we presented a variable-invariant two-dimensional state space model for multivariate time series. By replacing sequential updates along the variable axis with a global, permutation-invariant descriptor, our approach eliminates ordering dependence while preserving global inter-variable correlations and enabling full parallelism. To further enhance representation capacity, we introduced a multi-branch design that integrates long- and short-horizon temporal pathways with a dedicated frequency-domain pathway, adaptively fused through a lightweight gating mechanism.

Extensive experiments on forecasting, classification, and anomaly detection benchmarks demonstrate that our model consistently outperforms state-of-the-art baselines in both accuracy and efficiency. Comprehensive ablation and case studies confirm the robustness of the variable-invariant formulation and highlight the complementary benefits of the multi-branch architecture. These results establish our approach as a scalable and general framework for modeling complex temporal–spectral patterns in multivariate time series.

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

## A   RELATED WORKS

### A.1   NEURAL MODELS FOR TIME SERIES MODELING

Deep learning approaches for multivariate time series have evolved from architectures designed to capture temporal evolution to models that also account for complex cross-variable dependencies (Qiu et al., 2025). Early methods based on CNNs and RNNs effectively extracted local temporal patterns and autoregressive dynamics, yielding strong performance on short- to medium-range horizons (Lai et al., 2018; Shen et al., 2020; Franceschi et al., 2019; He & Zhao, 2019). Subsequent Transformer-based architectures advanced the field by leveraging self-attention and task-specific inductive biases to model global temporal context, demonstrating notable improvements across various tasks (Liu et al., 2023; Nie et al., 2022; Zhou et al., 2022). However, the quadratic scaling of self-attention with respect to sequence length severely limits scalability on long sequences, and this challenge persists despite efforts to mitigate it via sparsification or kernelization strategies (Keles et al., 2023). More recently, SSMs (Wang et al., 2025; Patro & Agneeswaran, 2024) and convolutional alternatives (Luo & Wang, 2024) have emerged as efficient solutions, offering linear or near-linear complexity with accuracy competitive to that of Transformers. Nevertheless, most of these models are inherently one-dimensional, evolving only along the temporal axis, and thus remain limited in their ability to capture dynamic inter-variable dependencies that are critical in multivariate time series.

### A.2   STATE SPACE MODELS AND 2D EXTENSIONS

Efforts to achieve high performance while preserving linear computational complexity have led to the development of SSM–based methods. Modern 1D SSMs such as S4 (Gu et al., 2021) and Mamba (Gu & Dao, 2023) leverage structured recurrences and input-conditioned parameters (*e.g.*, selective updates), achieving accuracy competitive with or superior to attention mechanisms on long sequences while retaining favorable scaling. However, because these formulations evolve only along the temporal axis, they are limited in their ability to simultaneously capture time-varying cross-variable dependencies. This limitation becomes particularly salient in domains such as vision and multivariate time series, where correlations among variables are as critical as temporal dynamics.

To address this, various extensions of Mamba have been proposed. For example, TimePro (Ma et al.), CMamba (Zeng et al., 2024), Simba (Patro & Agneeswaran, 2024), ET-Mamba (Jeong et al., 2025), and GrootV-T (Xiao et al., 2024) introduce mechanisms for modeling variable-level structure or topological priors. Yet, these approaches remain insufficient to fundamentally resolve the challenge, as they often rely on heuristics or task-specific inductive biases. Consequently, the need for two-dimensional SSMs has become increasingly apparent. Building on classical multidimensional formulations such as Roesser's model (Kung et al., 1977), recent work has extended 2D SSMs into deep learning (Baron et al., 2024; Zhang et al., 2025; Behrouz et al., 2024). Among these, Chimera (Behrouz et al., 2024) instantiates a 2D Mamba architecture with coupled temporal and variable dynamics, thereby enabling richer cross-variable interactions for time series data. Nonetheless, because SSMs are intrinsically causal along their scan axis, Chimera inherits a dependence on variable ordering and requires sequential scans over the variable dimension. This not only limits permutation robustness but also restricts parallelism as the number of variables grows.

To overcome these challenges, we propose a novel variable-invariant 2D SSM that resolves ordering dependencies through a global, permutation-invariant mechanism, enabling more faithful modeling of multivariate correlations while retaining computational efficiency.

## B   VARIABLE-INVARIANT 2D SSM

### B.1   DISCRETIZATION OF THE 2D SSM

The continuous our 2D SSM is given by:

$$\frac{d}{dt} \begin{bmatrix} h_h(t,c) \\ h_v(t,c) \end{bmatrix} = \begin{bmatrix} A_h & 0 \\ A_{vh} & A_v \end{bmatrix} \begin{bmatrix} h_h(t,c) \\ h_v(t,c) \end{bmatrix} + \begin{bmatrix} A_{h\psi} \\ A_{v\psi} \end{bmatrix} \psi(t) + \begin{bmatrix} B_h \\ B_v \end{bmatrix} x(t,c).$$

In matrix form:

$$\mathcal{A} = \begin{bmatrix} A_h & 0 \\ A_{vh} & A_v \end{bmatrix}, \quad \mathcal{B} = \begin{bmatrix} A_{h\psi} & B_h \\ A_{v\psi} & B_v \end{bmatrix}.$$

The general solution of the continuous state equation is:

$$\mathbf{h}(t) = e^{\mathcal{A}(t-t_0)}\mathbf{h}(t_0) + \int_{t_0}^{t} e^{\mathcal{A}(t-\tau)}\mathcal{B}\mathbf{u}(\tau)d\tau,$$

where $\mathbf{h}(t) = \begin{bmatrix} h_h(t, c) \\ h_v(t, c) \end{bmatrix}$ and $\mathbf{u}(t) = \begin{bmatrix} \psi(t) \\ x(t, c) \end{bmatrix}$.

Under the ZOH assumption, when $t_0 = k\Delta$ and $t = (k+1)\Delta$:

$$\mathbf{h}((k+1)\Delta) = e^{\mathcal{A}\Delta}\mathbf{h}(k\Delta) + \int_{k\Delta}^{(k+1)\Delta} e^{\mathcal{A}((k+1)\Delta-\tau)}\mathcal{B}\mathbf{u}[k]d\tau.$$

Since $\mathbf{u}(\tau) = \mathbf{u}[k]$ is constant over the interval due to ZOH:

$$\mathbf{h}[k+1] = e^{\mathcal{A}\Delta}\mathbf{h}[k] + \left( \int_{k\Delta}^{(k+1)\Delta} e^{\mathcal{A}((k+1)\Delta-\tau)}\mathcal{B}d\tau \right)\mathbf{u}[k]$$

$$= e^{\mathcal{A}\Delta}\mathbf{h}[k] + \left( \int_{0}^{\Delta} e^{\mathcal{A}(\Delta-\lambda)}\mathcal{B}d\lambda \right)\mathbf{u}[k]$$

$$= e^{\mathcal{A}\Delta}\mathbf{h}[k] + \left( \int_{0}^{\Delta} e^{\mathcal{A}\nu}\mathcal{B}d\nu \right)\mathbf{u}[k].$$

Therefore, the discretized system is:

$$\mathbf{h}[k+1, c] = \bar{\mathcal{A}}\mathbf{h}[k, c] + \bar{\mathcal{B}}\begin{bmatrix} \psi[k] \\ x[k, c] \end{bmatrix},$$

where:

$$\bar{\mathcal{A}} = e^{\mathcal{A}\Delta} = \exp\left( \begin{bmatrix} A_h & 0 \\ A_{vh} & A_v \end{bmatrix}\Delta \right),$$

$$\bar{\mathcal{B}} = \left( \int_{0}^{\Delta} e^{\mathcal{A}\tau}d\tau \right)\mathcal{B}.$$

### B.2 Convolution Formulation of Variable-Invariant 2D SSM

Applying the recurrent rules in Eq. 11- 13, we can write the output as:

$$y[t, c] = \sum_{\tau}^{t-1} K_\psi[t-1-\tau]\psi[\tau] + \sum_{\tau}^{t-1} K_x[t-1-\tau]x[\tau, c],$$

where global info kernel $K_\psi[k] = [C_h, C_v]\bar{\mathcal{A}}_k\bar{\mathcal{B}}[:, 0]$ and input kernel $K_x[k] = [C_h, C_v]\bar{\mathcal{A}}_k\bar{\mathcal{B}}[:, 1]$.

## C Theoretical Proof

### C.1 Proposition 4.1

*Let $\pi$ be any permutation of $\{1, \ldots, C\}$ and define $\psi_\pi(t) := \phi\big(\{W_v h_v(t, \pi(c))\}_{c=1}^{C}\big)$. Assume (i) $W_v$ is variable-shared (independent of c), and (ii) $\phi$ is permutation-invariant on multisets, i.e., $\phi(\{z_c\}_{c=1}^{C}) = \phi(\{z_{\pi(c)}\}_{c=1}^{C})$ for any collection $\{z_c\}$. Then $\psi_\pi(t) = \psi(t)$ for all t, where $\psi(t) = \phi(\{W_v h_v(t, c)\}_{c=1}^{C})$.*

*Proof.* Define the variable-wise set $S_t = \{z_c(t)\}_{c=1}^C$ with $z_c(t) = W_v h_v(t, c)$. By definition, $\psi(t) = \phi(S_t)$. Under permutation $\pi$, the collection becomes

$$S_t^\pi = \{z_{\pi(c)}(t)\}_{c=1}^C = \{W_v h_v(t, \pi(c))\}_{c=1}^C.$$

Since $\phi$ is permutation-invariant, $\phi(S_t^\pi) = \phi(S_t)$. Hence $\psi_\pi(t) = \psi(t)$ for all $t$. $\qquad\square$

## C.2 PROPOSITION 4.2

*Assume $A_h, A_v, A_{h\psi}, A_{v\psi}, A_{vh}, B_h, B_v$ are variable-shared (independent of c), and let $\psi(t)$ be defined as above with a permutation-invariant $\phi$ (Prop. 4.1). For any permutation $\pi$ of $\{1, \ldots, C\}$, consider the permuted inputs $x^\pi(t, c) := x(t, \pi(c))$ and permuted initial states $h^\pi(0, c) := h(0, \pi(c))$. Then the unique solutions to Eq. 8-9 satisfy $h_h^\pi(t, c) = h_h(t, \pi(c))$, $h_v^\pi(t, c) = h_v(t, \pi(c)) \ \forall \ t, c$, i.e., the dynamics are permutation equivariant with respect to the variable axis.*

*Proof.* Let $h_h, h_v$ be a solution of the continuous dynamics

$$\frac{\partial h_h(t, c)}{\partial t} = A_h h_h(t, c) + A_{h\psi}\psi(t) + B_h x(t, c),$$

$$\frac{\partial h_v(t, c)}{\partial t} = A_v h_v(t, c) + A_{v\psi}\psi(t) + A_{vh} h_h(t, c) + B_v x(t, c),$$

with initial conditions $h_h(0, c), h_v(0, c)$. Here, the matrices $A_h, A_v, A_{h\psi}, A_{v\psi}, B_{vh}, B_h, B_v$ are *variable-shared* (do not depend on c), and $\psi(t) = \phi(\{W_v h_v(t, c)\}_{c=1}^C)$ with $\phi$ permutation-invariant (Prop. 4.1). Fix any permutation $\pi$ of $\{1, \ldots, C\}$ and let the permuted inputs $x^\pi(t, c) := x(t, \pi(c))$, and the permuted initial states $h_h^\pi(0, c) := h_h(0, \pi(c)), h_v(0, c) := h_v(0, \pi(c))$. Consider the $\tilde{h}_h(t, c) := h_h(t, \pi(c)), \tilde{h}_v(t, c) := h_v(t, \pi(c))$. We show that $(\tilde{h}_h, \tilde{h}_v)$ satisfy the permuted system driven by $x^\pi$ with initial conditions $h_h^\pi(0, \cdot), h_v^\pi(0, \cdot)$.

**$\psi$ is invariant under variable permutations.**

By Prop. 4.1, for any $t$, $\psi^\pi(t) := \psi(t)$. Hence, the pooled descriptor used in the dynamics is unchanged by the permutation.

**$\tilde{h}_h, \tilde{h}_v$ satisfy the permuted ODEs.**

Differentiating $\tilde{h}_h$ and using Eq. 8,

$$\begin{aligned}
\frac{\partial}{\partial t}\tilde{h}_h(t, c) &= \frac{\partial}{\partial t}h_h(t, \pi(c)) \\
&= A_h h_h(t, \pi(c)) + A_{h\psi}\psi(t) + B_h x(t, \pi(c)) \\
&= A_h \tilde{h}_h(t, c) + A_{h\psi}\psi(t) + B_h x^\pi(t, c).
\end{aligned}$$

Likewise, using Eq. 9,

$$\frac{\partial}{\partial t}\tilde{h}_v(t, c) = A_v \tilde{h}_v(t, c) + A_{v\psi}\psi(t) + A_{vh}\tilde{h}_h(t, c) + B_v x^\pi(t, c).$$

By construction, $\tilde{h}_h(0, c) = h_h^\pi(0, c)$ and $\tilde{h}_v(0, c) = h_v^\pi(0, c)$. Thus, $(\tilde{h}_h, \tilde{h}_v)$ are the solution of the permuted system with input $x^\pi$ and the permuted initial states.

**Uniqueness**

The systems are linear time-varying (through $\psi$) but globally Lipschitz in the states. Hence, the solutions are unique. Therefore, the unique solution $(h_h^\pi, h_v^\pi)$ of the permuted system coincides with $(\tilde{h}_h, \tilde{h}_v)$, *i.e.*,

$$(h_h^\pi(t, c), h_v^\pi(t, c)) = (\tilde{h}_h(t, c), \tilde{h}_v(t, c)) = (h_h(t, \pi(c)), h_v(t, \pi(c))), \quad \forall t, c.$$

This proves permutation *equivariance* of the dynamics with respect to the variable axis. $\qquad\square$

# D  ALGORITHMS

For clarity, we denote variable-axis updates using a for-loop. However, unlike prior 2D SSMs that require sequential scans across variables, all updates for different variables at time $t$ are conditionally independent given the pooled descriptor $\psi$. Thus, these operations can be executed fully in parallel across the variable axis. The only additional cost is an $O(C)$ pooling step per time slice, which corresponds to a parallel aggregation (*e.g.*, mean, gated pooling, or attention) over all variables. Consequently, the overall complexity reduces to a single temporal selective scan combined with parallel variable-axis updates, ensuring both scalability and permutation invariance.

---

**Algorithm 1:** Variable-Invariant 2D SSM (Forward)

---

**Input**  : $X \in \mathbb{R}^{C \times T}$    (Inputs for $C$ variables over $T$ steps)
**Input**  : $\phi$   (Permutation-invariant aggregator)
**Output:** $Y$

1  Initialize $h_h$, $h_v$
2  **for** $t \leftarrow 1$ **to** $T$ **do**
3     // single temporal scan
4     **for** $c \leftarrow 1$ **to** $C$ **do**
5         $z_c(t) \leftarrow W_v\, h_v[t-1, c]$
6     $\psi[t] \leftarrow \phi\big(\{z_c(t)\}_{c=1}^{C}\big)$
7     **for** $c \leftarrow 1$ **to** $C$ **do**
8         $h_h[t, c] \leftarrow \bar{A}_h h_h[t-1, c] + \bar{B}_h^x X[t, c] + \bar{B}_h^\psi \psi[t]$
9         $h_v[t, c] \leftarrow \bar{A}_v h_v[t-1, c] + \bar{A}_{vh} h_h[t, c] + \bar{B}_v^x X[t, c] + \bar{B}_v^\psi \psi[t]$
10 $Y \leftarrow C\,[\,h_h, h_v\,]$
11 **return** $Y$

---

---

**Algorithm 2:** Variable-Invariant 2D Mamba Block

---

**Input**  : $X \in \mathbb{R}^{C \times T}$, aggregator $\phi$
**Input**  : Step sizes $\Delta_{\text{long}}, \Delta_{\text{short}}, \Delta_{\text{freq}}$
**Output:** $H$

1  $H_{\text{long}} \leftarrow \text{VI2DSSMFORWARD}(X, \phi; \Delta = \Delta_{\text{long}})$
2  $H_{\text{short}} \leftarrow \text{VI2DSSMFORWARD}(X, \phi; \Delta = \Delta_{\text{short}})$
3  $X_f \leftarrow \text{FFT}(X); \quad H_{\text{freq}} \leftarrow \text{VI2DSSMFORWARD}(X_f, \phi; \Delta = \Delta_{\text{freq}})$
4  $G \leftarrow \sigma\big(W_g\,[\,\text{POOL}(H_{\text{long}}) \,\|\, \text{POOL}(H_{\text{short}}) \,\|\, \text{POOL}(H_{\text{freq}})\,]\big)$
5  $H \leftarrow G_1 \odot H_{\text{long}} + G_2 \odot H_{\text{short}} + G_3 \odot H_{\text{freq}}$
6  **return** $H$

---

# E  AGGREGATION FUNCTION $\psi$ AND COMPUTATIONAL BEHAVIOR

## E.1  CHOICE OF THE AGGREGATION FUNCTION $\psi$

The proposed formulation allows any permutation-invariant aggregation operator $\psi$, such as mean/sum pooling or attention-based pooling. In practice, the default choice of $\psi$ is selected according to the inductive bias and computational structure of each task:

- **Forecasting.** Multi-step forecasting tasks (*e.g.*, ETTh1/ETTm1) involve strong shared seasonal components across variables and are highly sensitive to variance amplification. Mean pooling produces a low-variance global descriptor that suppresses channel-wise noise and stabilizes long-horizon prediction. This property leads to more robust extrapolation than attention-based pooling.

- **Classification & Anomaly Detection.** In contrast, these tasks rely on discriminative, potentially sparse cross-variable interactions, where only a subset of channels may indicate

class identity or abnormal behavior. Mean pooling may dilute these informative variables, whereas attention pooling selectively emphasizes salient channels, improving representation quality.

### E.2 GPU Throughput and Memory Usage

To clarify the computational implications of different choices of $\psi$, Table 6 summarizes the work complexity, parallel span, and peak memory usage. Let $C$ denote the number of variables and $d$ denote the hidden dimension.

Table 6: Computational analysis of permutation-invariant aggregation functions. The trade-off between representational expressiveness and memory/compute overhead motivates task-adaptive selection of $\psi$.

| Aggregation $\psi$ | Work (per timestep) | Parallel Span | Peak Memory Overhead | Characteristics |
|---|---|---|---|---|
| Mean | $O(Cd)$ | $O(\log C)$ | $O(d)$ | Stable, low-variance descriptor |
| Sum | $O(Cd)$ | $O(\log C)$ | $O(d)$ | Same cost as mean, without normalization |
| Attention | $O(Cd^2)$ | $O(\log C)$ | $O(Cd)$ | Expressive, selectively emphasizes informative variables |

Mean/sum pooling require only a single accumulator vector and do not store intermediate key/value tensors, resulting in minimal peak memory. Attention pooling, however, stores per-variable projections and attention scores, causing memory to grow linearly with $C$. This explains why forecasting, which benefits from stability and scalability, is paired with mean pooling, whereas expressive tasks (classification/anomaly detection) are paired with attention-based pooling.

### E.3 Computational Efficiency Analysis

We provide a quantitative comparison of computational cost across representative transformer- and SSM-based multivariate forecasting models. All methods were evaluated under the same conditions: ECL dataset, input window length $L = 96$, prediction window $H = 720$, and batch size 16. We report measurements only for models with official code enabling reproducible FLOPs and peak memory profiling.

Table 7: Measured peak GPU memory and forward FLOPs.

| Method | Memory (MB) | FLOPs (G) |
|---|---|---|
| Ours | 546.62 | 11.99 |
| TimePro | 164.35 | 35.04 |
| S-Mamba | 270.76 | 96.13 |
| PatchTST | 675.26 | 85.64 |
| iTransformer | 1155.14 | 73.20 |

A conventional 2D State Space Model processes dynamics along both temporal and variable axes. Let $L$ be the sequence length, $C$ the number of variables, and $d$ the hidden dimension. The standard cost is:

$$\underbrace{O(LCd)}_{\text{temporal SSM scan}} + \underbrace{O(LCd)}_{\text{variable-axis recursion}} = O(2LCd).$$

The second term is sequential in $C$, restricting GPU parallelism.

We replace the recursive variable update with a single permutation-invariant pooled descriptor:

$$\underbrace{O(LCd)}_{\text{temporal SSM scan}} + \underbrace{O(Cd)}_{\text{Global pooling}} = O(LCd) + O(Cd).$$

Crucially, the additional $O(Cd)$ term is computed in one fully parallel GPU kernel, rather than a sequential pass. This reduces practical computation to a single directional SSM scan, eliminating any

dependence on variable ordering or recursive updates. Because our model does not evolve states along the variable axis, it avoids the second $O(LCd)$ recurrence found in SSM baselines (Time-Pro, S-Mamba). The measured FLOPs are therefore markedly lower, despite additional frequency-domain modeling.

## F  EXPERIMENTS DETAILS

To assess the effectiveness of the proposed method, we conducted extensive comparisons against a broad range of state-of-the-art models, including recent Mamba-based approaches. Baseline results were obtained from the corresponding literature to ensure fairness and consistency. The comparison models include Chimera (Behrouz et al., 2024), TimePro (Ma et al.), Simba (Patro & Agneeswaran, 2024), TCN (Luo & Wang, 2024), iTransformer (Liu et al., 2023), RLinear (Li et al., 2023), PatchTST (Nie et al., 2022), Crossformer (Zhang & Yan, 2023), TiDE (Das et al., 2023), Times-Net (Wu et al., 2022a), DLinear (Zeng et al., 2023), SCINet (Liu et al., 2022a), FEDformer (Zhou et al., 2022), Stationary (Liu et al., 2022b), N-HiTS (Challu et al., 2023), N-BEATS (Oreshkin et al., 2019), ETSformer (Woo et al., 2022), LightTS (Zhang et al.), Autoformer (Wu et al., 2021), Pyraformer (Liu et al.), Informer (Zhou et al., 2021), Reformer (Kitaev et al., 2020), LSTM (Hochreiter & Schmidhuber, 1997), LSTNet (Lai et al., 2018), LSSL (Gu et al., 2021), Flowformer (Wu et al., 2022b) and LogTrans (Li et al., 2019).

Table 8: Data description for time series forecasting.

| Dataset | Types | Sample Number (train/validation/test) | Variable Dimension | Prediction Length |
|---|---|---|---|---|
| ETTh1, ETTh2 | | (8545/2881/2881) | 7 | {96 192, 336, 720} |
| ETTm1, ETTm2 | | (34465/11521/11521) | 7 | {96 192, 336, 720} |
| Electricity | | (18317/2633/5261) | 321 | {96 192, 336, 720} |
| Traffic | Long-term | (12185/1757/3509) | 862 | {96 192, 336, 720} |
| Weather | | (36792/5271/10540) | 21 | {96 192, 336, 720} |
| Exchange | | (5120/665/1422) | 8 | {96 192, 336, 720} |
| M4-Yearly | | (23000/0/23000) | 1 | 6 |
| M4-Quarterly | | (24000/0/24000) | 1 | 8 |
| M4-Monthly | | (48000/0/48000) | 1 | 18 |
| M4-Weekly | Short-term | (359/0/359) | 1 | 13 |
| M4-Daily | | (4227/0/4227) | 1 | 14 |
| M4-Hourly | | (414/0/414) | 1 | 48 |

### F.1  DATASETS

**Forecasting.**   We used eight benchmark datasets for long-term forecasting (Weather, Traffic, Electricity, Exchange, and four ETT datasets (Zhou et al., 2021)) and the M4 dataset for short-term forecasting (Makridakis et al., 2018). Long-term datasets consist of single continuous sequences with samples generated by sliding windows, whereas M4 contains 100,000 heterogeneous series from diverse domains. Dataset descriptions are provided in Table 8.

**Classification & Anomaly Detection.**   For classification, we used ten benchmark datasets from the UEA archive (Bagnall et al., 2018), spanning diverse domains such as vision, audio, industry monitoring, and medical diagnosis, with most tasks involving around ten classes. For anomaly detection, we adopted five widely used benchmarks: SMD (Su et al., 2019), SWaT (Mathur & Tippenhauer, 2016), PSM (Abdulaal et al., 2021), MSL, and SMAP (Hundman et al., 2018), covering domains such as server machines, spacecraft, and critical infrastructure. Dataset descriptions are summarized in Table 9.

Table 9: Data description for time series classification and anomaly detection.

| Dataset | Types | Sample Number (train/validation/test) | Variable Dimension | Series Length (classification) Sliding Window Length (anomaly detection) |
|---|---|---|---|---|
| EthanolConcentration | | (261/-/263) | 3 | 1751 |
| FaceDetection | | (5890/-/3524) | 144 | 62 |
| Handwriting | | (150/-/850) | 3 | 152 |
| Heartbeat | | (204/-/205) | 61 | 405 |
| JapaneseVowels | Classification | (270/-/370) | 12 | 29 |
| PEMS-SF | | (267/-/173) | 963 | 144 |
| SelfRegulationSCP1 | | (268/-/293) | 6 | 896 |
| SelfRegulationSCP2 | | (200/-/180) | 7 | 1152 |
| SpokenArabicDigits | | (6599/-/2199) | 13 | 93 |
| UWaveGestureLibrary | | (120/-/320) | 3 | 315 |
| SMD | | (566724/141681/708420) | 38 | 100 |
| MSL | | (44653/11664/73729) | 55 | 100 |
| SMAP | Anomaly Detection | (108146/27037/427617) | 25 | 100 |
| SWaT | | (396000/99000/449919) | 51 | 100 |
| PSM | | (105984/26497/87841) | 25 | 100 |

## F.2 Full Experimental Results

The overall results for long- and short-term forecasting, classification, and anomaly detection are reported in Table 10-13.

Table 10: Full results of long-term time-series forecasting.

| | | Ours | | Chimera | | TimePro | | Simba | | TCN | | iTransformer | | RLinear | | PatchTST | | Crossformer | | TiDE | | TimesNet | | DLinear | | SCINet | | FEDformer | |
|---|---|---|---|---|---|---|---|---|---|---|---|---|---|---|---|---|---|---|---|---|---|---|---|---|---|---|---|---|---|
| | | MSE | MAE | MSE | MAE | MSE | MAE | MSE | MAE | MSE | MAE | MSE | MAE | MSE | MAE | MSE | MAE | MSE | MAE | MSE | MAE | MSE | MAE | MSE | MAE | MSE | MAE | MSE | MAE |
| ETTm1 | 96 | **0.287** | **0.342** | 0.318 | 0.354 | 0.326 | 0.364 | 0.324 | 0.360 | 0.292 | 0.346 | 0.334 | 0.368 | 0.355 | 0.376 | 0.329 | 0.367 | 0.404 | 0.426 | 0.364 | 0.387 | 0.338 | 0.375 | 0.345 | 0.372 | 0.418 | 0.438 | 0.379 | 0.419 |
| | 192 | **0.326** | **0.367** | 0.331 | 0.369 | 0.367 | 0.383 | 0.363 | 0.382 | 0.332 | 0.368 | 0.377 | 0.391 | 0.391 | 0.392 | 0.367 | 0.385 | 0.450 | 0.451 | 0.398 | 0.404 | 0.374 | 0.387 | 0.380 | 0.389 | 0.439 | 0.450 | 0.426 | 0.441 |
| | 336 | **0.359** | **0.387** | 0.363 | 0.389 | 0.402 | 0.409 | 0.395 | 0.405 | 0.365 | 0.391 | 0.426 | 0.420 | 0.424 | 0.415 | 0.399 | 0.410 | 0.532 | 0.515 | 0.428 | 0.425 | 0.410 | 0.411 | 0.413 | 0.413 | 0.490 | 0.485 | 0.445 | 0.459 |
| | 720 | **0.409** | 0.416 | **0.409** | 0.415 | 0.469 | 0.446 | 0.451 | 0.437 | 0.416 | 0.417 | 0.491 | 0.459 | 0.487 | 0.450 | 0.454 | 0.439 | 0.666 | 0.589 | 0.487 | 0.461 | 0.478 | 0.450 | 0.474 | 0.453 | 0.595 | 0.550 | 0.543 | 0.490 |
| | Avg | **0.345** | **0.378** | 0.355 | 0.381 | 0.391 | 0.400 | 0.383 | 0.396 | 0.351 | 0.381 | 0.407 | 0.410 | 0.414 | 0.407 | 0.387 | 0.400 | 0.513 | 0.496 | 0.419 | 0.419 | 0.400 | 0.406 | 0.403 | 0.407 | 0.485 | 0.481 | 0.448 | 0.452 |
| ETTm2 | 96 | 0.169 | 0.259 | 0.169 | 0.265 | 0.178 | 0.260 | 0.177 | 0.263 | **0.166** | **0.256** | 0.180 | 0.264 | 0.182 | 0.265 | 0.175 | 0.259 | 0.287 | 0.366 | 0.207 | 0.305 | 0.187 | 0.267 | 0.193 | 0.292 | 0.286 | 0.377 | 0.203 | 0.287 |
| | 192 | 0.223 | 0.295 | **0.221** | **0.290** | 0.242 | 0.303 | 0.245 | 0.306 | 0.222 | 0.293 | 0.250 | 0.309 | 0.246 | 0.304 | 0.241 | 0.302 | 0.414 | 0.492 | 0.290 | 0.364 | 0.249 | 0.309 | 0.284 | 0.362 | 0.399 | 0.445 | 0.269 | 0.328 |
| | 336 | 0.279 | 0.332 | 0.279 | 0.339 | 0.303 | 0.342 | 0.304 | 0.343 | **0.272** | **0.324** | 0.311 | 0.348 | 0.307 | 0.342 | 0.305 | 0.343 | 0.597 | 0.542 | 0.377 | 0.422 | 0.321 | 0.351 | 0.369 | 0.427 | 0.637 | 0.591 | 0.325 | 0.366 |
| | 720 | 0.362 | 0.385 | 0.342 | 0.376 | 0.400 | 0.399 | 0.400 | 0.399 | 0.351 | 0.381 | 0.412 | 0.407 | 0.407 | 0.398 | 0.402 | 0.400 | 1.730 | 1.042 | 0.558 | 0.524 | 0.408 | 0.403 | 0.554 | 0.522 | 0.960 | 0.735 | 0.421 | 0.415 |
| | Avg | 0.258 | 0.318 | **0.252** | **0.317** | 0.281 | 0.326 | 0.271 | 0.327 | 0.253 | 0.314 | 0.288 | 0.332 | 0.286 | 0.327 | 0.281 | 0.326 | 0.757 | 0.610 | 0.358 | 0.404 | 0.291 | 0.333 | 0.350 | 0.401 | 0.571 | 0.537 | 0.305 | 0.349 |
| ETTh1 | 96 | **0.365** | 0.394 | 0.366 | 0.392 | 0.375 | 0.398 | 0.379 | 0.395 | 0.368 | 0.394 | 0.386 | 0.405 | 0.386 | 0.400 | 0.414 | 0.419 | 0.423 | 0.448 | 0.479 | 0.464 | 0.384 | 0.402 | 0.386 | 0.400 | 0.654 | 0.599 | 0.376 | 0.419 |
| | 192 | 0.407 | 0.417 | **0.402** | 0.414 | 0.427 | 0.429 | 0.432 | 0.424 | 0.405 | **0.413** | 0.441 | 0.436 | 0.437 | 0.424 | 0.460 | 0.445 | 0.471 | 0.474 | 0.525 | 0.492 | 0.436 | 0.429 | 0.437 | 0.432 | 0.719 | 0.631 | 0.420 | 0.448 |
| | 336 | **0.385** | 0.416 | 0.406 | 0.419 | 0.472 | 0.450 | 0.473 | 0.443 | 0.391 | **0.412** | 0.487 | 0.458 | 0.479 | 0.446 | 0.501 | 0.466 | 0.570 | 0.546 | 0.565 | 0.515 | 0.491 | 0.469 | 0.481 | 0.459 | 0.778 | 0.659 | 0.459 | 0.465 |
| | 720 | **0.434** | **0.452** | 0.458 | 0.477 | 0.476 | 0.474 | 0.483 | 0.469 | 0.450 | 0.461 | 0.503 | 0.491 | 0.481 | 0.470 | 0.500 | 0.488 | 0.653 | 0.621 | 0.594 | 0.558 | 0.521 | 0.500 | 0.519 | 0.516 | 0.836 | 0.699 | 0.506 | 0.507 |
| | Avg | **0.397** | **0.419** | 0.408 | 0.425 | 0.438 | 0.438 | 0.441 | 0.432 | 0.404 | 0.420 | 0.454 | 0.447 | 0.446 | 0.434 | 0.469 | 0.454 | 0.529 | 0.522 | 0.541 | 0.507 | 0.458 | 0.450 | 0.456 | 0.452 | 0.747 | 0.647 | 0.440 | 0.460 |
| ETTh2 | 96 | **0.257** | 0.329 | 0.262 | 0.327 | 0.293 | 0.345 | 0.290 | 0.339 | 0.263 | 0.332 | 0.297 | 0.349 | 0.288 | 0.338 | 0.302 | 0.348 | 0.745 | 0.584 | 0.400 | 0.440 | 0.340 | 0.374 | 0.333 | 0.387 | 0.707 | 0.621 | 0.358 | 0.397 |
| | 192 | **0.312** | **0.367** | 0.320 | 0.372 | 0.367 | 0.394 | 0.373 | 0.390 | 0.320 | 0.374 | 0.380 | 0.400 | 0.374 | 0.390 | 0.388 | 0.400 | 0.877 | 0.656 | 0.528 | 0.509 | 0.402 | 0.414 | 0.477 | 0.476 | 0.860 | 0.689 | 0.429 | 0.439 |
| | 336 | **0.303** | **0.367** | 0.316 | 0.381 | 0.419 | 0.431 | 0.419 | 0.426 | 0.313 | 0.376 | 0.428 | 0.432 | 0.415 | 0.426 | 0.426 | 0.433 | 1.043 | 0.731 | 0.643 | 0.571 | 0.452 | 0.452 | 0.594 | 0.541 | 1.000 | 0.744 | 0.496 | 0.487 |
| | 720 | **0.381** | **0.425** | 0.389 | 0.430 | 0.427 | 0.445 | 0.407 | 0.431 | 0.392 | 0.433 | 0.427 | 0.445 | 0.420 | 0.440 | 0.431 | 0.446 | 1.104 | 0.763 | 0.874 | 0.679 | 0.462 | 0.468 | 0.831 | 0.657 | 1.249 | 0.838 | 0.463 | 0.474 |
| | Avg | **0.313** | **0.372** | 0.321 | 0.377 | 0.377 | 0.403 | 0.361 | 0.391 | 0.322 | 0.379 | 0.383 | 0.407 | 0.374 | 0.398 | 0.387 | 0.407 | 0.942 | 0.684 | 0.611 | 0.550 | 0.414 | 0.427 | 0.559 | 0.515 | 0.954 | 0.723 | 0.437 | 0.449 |
| ECL | 96 | **0.127** | **0.223** | 0.132 | 0.234 | 0.139 | 0.234 | 0.165 | 0.253 | 0.129 | 0.226 | 0.148 | 0.240 | 0.201 | 0.281 | 0.181 | 0.270 | 0.219 | 0.314 | 0.237 | 0.329 | 0.168 | 0.272 | 0.197 | 0.282 | 0.247 | 0.345 | 0.193 | 0.308 |
| | 192 | 0.147 | 0.234 | 0.144 | **0.223** | 0.156 | 0.249 | 0.173 | 0.262 | **0.143** | 0.239 | 0.162 | 0.253 | 0.201 | 0.283 | 0.188 | 0.274 | 0.231 | 0.322 | 0.236 | 0.330 | 0.184 | 0.289 | 0.196 | 0.285 | 0.257 | 0.355 | 0.201 | 0.315 |
| | 336 | 0.165 | 0.266 | **0.156** | 0.259 | 0.172 | 0.267 | 0.188 | 0.277 | 0.161 | **0.259** | 0.178 | 0.269 | 0.215 | 0.298 | 0.204 | 0.293 | 0.246 | 0.337 | 0.249 | 0.344 | 0.198 | 0.300 | 0.209 | 0.301 | 0.269 | 0.369 | 0.214 | 0.329 |
| | 720 | 0.196 | 0.295 | **0.184** | **0.280** | 0.209 | 0.299 | 0.214 | 0.305 | 0.191 | 0.286 | 0.225 | 0.317 | 0.257 | 0.331 | 0.246 | 0.324 | 0.280 | 0.363 | 0.284 | 0.373 | 0.220 | 0.320 | 0.245 | 0.333 | 0.299 | 0.390 | 0.246 | 0.355 |
| | Avg | 0.158 | 0.254 | **0.154** | **0.249** | 0.169 | 0.262 | 0.185 | 0.274 | 0.156 | 0.253 | 0.178 | 0.270 | 0.219 | 0.298 | 0.205 | 0.290 | 0.244 | 0.334 | 0.251 | 0.344 | 0.192 | 0.295 | 0.212 | 0.300 | 0.268 | 0.365 | 0.214 | 0.327 |
| Exchange | 96 | 0.085 | 0.204 | **0.077** | 0.198 | 0.085 | 0.204 | - | - | 0.080 | **0.196** | 0.086 | 0.206 | 0.093 | 0.217 | 0.088 | 0.205 | 0.256 | 0.367 | 0.094 | 0.218 | 0.107 | 0.234 | 0.088 | 0.218 | 0.267 | 0.396 | 0.148 | 0.278 |
| | 192 | 0.179 | 0.304 | **0.159** | 0.270 | 0.178 | 0.299 | - | - | 0.166 | 0.288 | 0.177 | 0.299 | 0.184 | 0.307 | 0.176 | 0.299 | 0.470 | 0.509 | 0.184 | 0.307 | 0.226 | 0.344 | 0.176 | 0.315 | 0.351 | 0.459 | 0.271 | 0.315 |
| | 336 | 0.330 | 0.415 | 0.311 | 0.344 | 0.328 | 0.414 | - | - | 0.307 | 0.398 | 0.331 | 0.417 | 0.351 | 0.432 | **0.301** | 0.397 | 1.268 | 0.883 | 0.349 | 0.431 | 0.367 | 0.448 | 0.313 | 0.427 | 1.324 | 0.853 | 0.460 | 0.427 |
| | 720 | **0.653** | **0.580** | 0.697 | 0.623 | 0.817 | 0.679 | - | - | 0.656 | 0.582 | 0.847 | 0.691 | 0.886 | 0.714 | 0.901 | 0.714 | 1.767 | 1.068 | 0.852 | 0.698 | 0.964 | 0.746 | 0.839 | 0.695 | 1.058 | 0.797 | 1.195 | 0.695 |
| | Avg | 0.311 | 0.375 | 0.311 | 0.358 | 0.352 | 0.399 | - | - | **0.302** | 0.366 | 0.360 | 0.403 | 0.378 | 0.417 | 0.367 | 0.404 | 0.940 | 0.707 | 0.370 | 0.413 | 0.416 | 0.443 | 0.354 | 0.414 | 0.750 | 0.626 | 0.519 | 0.429 |
| Traffic | 96 | 0.369 | 0.254 | **0.366** | 0.248 | - | - | 0.468 | 0.268 | 0.368 | 0.253 | 0.395 | 0.268 | 0.649 | 0.389 | 0.462 | 0.295 | 0.522 | 0.290 | 0.805 | 0.493 | 0.593 | 0.321 | 0.650 | 0.396 | 0.788 | 0.499 | 0.587 | 0.366 |
| | 192 | 0.381 | 0.278 | 0.394 | 0.292 | - | - | 0.413 | 0.317 | **0.379** | **0.261** | 0.417 | 0.276 | 0.601 | 0.366 | 0.466 | 0.296 | 0.530 | 0.293 | 0.756 | 0.474 | 0.617 | 0.336 | 0.598 | 0.370 | 0.789 | 0.505 | 0.604 | 0.373 |
| | 336 | **0.401** | 0.278 | 0.409 | 0.311 | - | - | 0.529 | 0.284 | 0.397 | **0.270** | 0.433 | 0.283 | 0.609 | 0.369 | 0.482 | 0.304 | 0.558 | 0.305 | 0.762 | 0.477 | 0.629 | 0.336 | 0.605 | 0.373 | 0.797 | 0.508 | 0.621 | 0.383 |
| | 720 | 0.444 | 0.295 | 0.443 | 0.294 | - | - | 0.564 | 0.297 | **0.440** | 0.296 | 0.467 | 0.302 | 0.647 | 0.387 | 0.514 | 0.322 | 0.589 | 0.328 | 0.719 | 0.449 | 0.640 | 0.350 | 0.645 | 0.394 | 0.841 | 0.523 | 0.626 | 0.382 |
| | Avg | 0.398 | 0.276 | 0.403 | 0.286 | - | - | 0.493 | 0.291 | **0.398** | **0.270** | 0.428 | 0.282 | 0.626 | 0.378 | 0.481 | 0.304 | 0.550 | 0.304 | 0.760 | 0.473 | 0.620 | 0.336 | 0.625 | 0.383 | 0.804 | 0.509 | 0.610 | 0.376 |
| Weather | 96 | 0.150 | 0.201 | **0.146** | 0.206 | 0.166 | 0.207 | 0.176 | 0.219 | 0.149 | **0.200** | 0.174 | 0.214 | 0.192 | 0.232 | 0.177 | 0.218 | 0.158 | 0.230 | 0.202 | 0.261 | 0.172 | 0.220 | 0.196 | 0.255 | 0.221 | 0.306 | 0.217 | 0.296 |
| | 192 | 0.198 | 0.245 | **0.189** | **0.239** | 0.216 | 0.254 | 0.222 | 0.260 | 0.196 | 0.245 | 0.221 | 0.254 | 0.240 | 0.271 | 0.225 | 0.259 | 0.206 | 0.277 | 0.242 | 0.298 | 0.219 | 0.261 | 0.237 | 0.296 | 0.261 | 0.340 | 0.276 | 0.336 |
| | 336 | 0.249 | 0.285 | 0.244 | 0.281 | 0.273 | 0.296 | 0.275 | 0.297 | **0.238** | **0.277** | 0.278 | 0.296 | 0.292 | 0.307 | 0.278 | 0.297 | 0.272 | 0.335 | 0.287 | 0.335 | 0.280 | 0.306 | 0.283 | 0.335 | 0.309 | 0.378 | 0.339 | 0.380 |
| | 720 | 0.312 | 0.326 | **0.297** | **0.309** | 0.351 | 0.346 | 0.350 | 0.349 | 0.314 | 0.334 | 0.358 | 0.347 | 0.364 | 0.353 | 0.354 | 0.348 | 0.398 | 0.418 | 0.351 | 0.386 | 0.365 | 0.359 | 0.345 | 0.381 | 0.377 | 0.427 | 0.403 | 0.428 |
| | Avg | 0.227 | 0.264 | **0.219** | **0.258** | 0.251 | 0.276 | 0.255 | 0.280 | 0.224 | 0.264 | 0.258 | 0.278 | 0.272 | 0.291 | 0.259 | 0.281 | 0.259 | 0.315 | 0.271 | 0.320 | 0.259 | 0.287 | 0.265 | 0.317 | 0.292 | 0.363 | 0.309 | 0.360 |

### F.2.1 Additional Analysis Results

Additional analysis with varying numbers of variables and sequence lengths is shown in Figure 5.

Table 11: Full results of short-term forecasting in the M4 dataset.

| | | Ours | Chimera | ModernTCN | PatchTST | TimesNet | N-HiTS | N-BEATS | ETSformer | LightTS | DLinear | FEDformer | Stationary | Autoformer | Pyraformer | Informer | Reformer |
|---|---|---|---|---|---|---|---|---|---|---|---|---|---|---|---|---|---|
| Yearly | SMAPE | 13.150 | **13.107** | 13.226 | 13.258 | 13.387 | 13.418 | 13.436 | 18.009 | 14.247 | 16.965 | 13.728 | 13.717 | 13.974 | 15.530 | 14.727 | 16.169 |
| | MASE | 2.950 | **2.902** | 2.957 | 2.985 | 2.996 | 3.045 | 3.043 | 4.487 | 3.109 | 4.283 | 3.048 | 3.078 | 3.134 | 3.711 | 3.418 | 3.800 |
| | OWA | 0.773 | **0.767** | 0.777 | 0.781 | 0.786 | 0.793 | 0.794 | 1.115 | 0.827 | 1.058 | 0.803 | 0.807 | 0.822 | 0.942 | 0.881 | 0.973 |
| Quarterly | SMAPE | 9.949 | **9.892** | 9.971 | 10.179 | 10.100 | 10.202 | 10.124 | 13.376 | 11.364 | 12.145 | 10.792 | 10.958 | 11.338 | 15.449 | 11.360 | 13.313 |
| | MASE | 1.160 | 1.105 | 1.167 | **0.803** | 1.182 | 1.194 | 1.169 | 1.906 | 1.328 | 1.520 | 1.283 | 1.325 | 1.365 | 2.350 | 1.401 | 1.775 |
| | OWA | 0.875 | 0.853 | 0.878 | **0.803** | 0.890 | 0.899 | 0.886 | 1.302 | 1.000 | 1.106 | 0.958 | 0.981 | 1.012 | 1.558 | 1.027 | 1.252 |
| Monthly | SMAPE | 12.563 | **12.549** | 12.556 | 12.641 | 12.670 | 12.791 | 12.677 | 14.588 | 14.014 | 13.514 | 14.260 | 13.917 | 13.958 | 17.642 | 14.062 | 20.128 |
| | MASE | 0.922 | **0.914** | 0.917 | 0.930 | 0.933 | 0.969 | 0.937 | 1.368 | 1.053 | 1.037 | 1.102 | 1.097 | 1.103 | 1.913 | 1.141 | 2.614 |
| | OWA | 0.869 | **0.864** | 0.866 | 0.876 | 0.878 | 0.899 | 0.880 | 1.149 | 0.981 | 0.956 | 1.012 | 0.998 | 1.002 | 1.511 | 1.024 | 1.927 |
| Others | SMAPE | 4.708 | **4.685** | 4.715 | 4.946 | 4.891 | 5.061 | 4.925 | 7.267 | 15.880 | 6.709 | 4.954 | 6.302 | 5.485 | 24.786 | 24.460 | 32.491 |
| | MASE | 3.165 | 3.007 | 3.107 | **2.985** | 3.302 | 3.216 | 3.391 | 5.240 | 11.434 | 4.953 | 3.264 | 4.064 | 3.865 | 18.581 | 20.960 | 33.355 |
| | OWA | 0.994 | 0.983 | **0.986** | 1.044 | 1.035 | 1.040 | 1.053 | 1.591 | 3.474 | 1.487 | 1.036 | 1.304 | 1.187 | 5.538 | 5.013 | 8.679 |
| Weighted Avg | SMAPE | 11.686 | **11.618** | 11.698 | 11.807 | 11.829 | 11.927 | 11.851 | 14.718 | 13.525 | 13.639 | 12.840 | 12.780 | 12.909 | 16.987 | 14.086 | 18.200 |
| | MASE | 1.549 | **1.528** | 1.556 | 1.590 | 1.585 | 1.613 | 1.599 | 2.408 | 2.111 | 2.095 | 1.701 | 1.756 | 1.771 | 3.265 | 2.718 | 4.223 |
| | OWA | 0.832 | **0.827** | 0.838 | 0.851 | 0.851 | 0.861 | 0.855 | 1.172 | 1.051 | 1.051 | 0.918 | 0.930 | 0.939 | 1.480 | 1.230 | 1.775 |

Table 12: Full results of the classification task (accuracy %)

| Datasets | LSTM | LSTNet | LSSL | Reformer | Informer | Pyraformer | Autoformer | Stationary | FEDformer | ETSformer | Flowformer | DLinear | LightTS | TimesNet | PatchTST | MTCN | Chimera | Ours |
|---|---|---|---|---|---|---|---|---|---|---|---|---|---|---|---|---|---|---|
| EthanolConcentration | 32.3 | 39.9 | 31.1 | 31.9 | 31.6 | 30.8 | 31.6 | 32.7 | 31.2 | 28.1 | 33.8 | 32.6 | 29.7 | 35.7 | 32.8 | 36.3 | 39.8 | 35.7 |
| FaceDetection | 57.7 | 65.7 | 66.7 | 68.6 | 67.0 | 65.7 | 68.4 | 68.0 | 66.0 | 66.3 | 67.6 | 68.0 | 67.5 | 68.6 | 68.3 | 70.8 | 70.4 | 69.6 |
| Handwriting | 15.2 | 25.8 | 24.6 | 27.4 | 32.8 | 29.4 | 36.7 | 31.6 | 28.0 | 32.5 | 33.8 | 27.0 | 26.1 | 32.1 | 29.6 | 30.6 | 32.9 | 35.3 |
| Heartbeat | 72.2 | 77.1 | 72.7 | 77.1 | 80.5 | 75.6 | 74.6 | 73.7 | 73.7 | 71.2 | 77.6 | 75.1 | 75.1 | 78.0 | 74.9 | 77.2 | 81.3 | 78.0 |
| JapaneseVowels | 79.7 | 98.1 | 98.4 | 97.8 | 98.9 | 98.4 | 96.2 | 99.2 | 98.4 | 95.9 | 98.9 | 96.2 | 96.2 | 98.4 | 97.5 | 98.8 | 99.1 | 98.4 |
| PEMS-SF | 39.9 | 86.7 | 86.1 | 82.7 | 81.5 | 83.2 | 82.7 | 87.3 | 80.9 | 86.0 | 83.8 | 75.1 | 88.4 | 89.6 | 89.3 | 89.1 | 89.5 | 88.1 |
| SelfRegulationSCP1 | 68.9 | 84.0 | 90.8 | 90.4 | 90.1 | 88.1 | 84.0 | 89.4 | 88.7 | 89.6 | 92.5 | 87.3 | 89.8 | 91.8 | 90.7 | 93.4 | 93.7 | 92.2 |
| SelfRegulationSCP2 | 46.6 | 52.8 | 52.2 | 56.7 | 53.3 | 53.3 | 50.6 | 57.2 | 54.4 | 55.0 | 56.1 | 50.5 | 51.1 | 57.2 | 57.8 | 60.3 | 59.9 | 57.8 |
| SpokenArabicDigits | 31.9 | 100.0 | 100.0 | 97.0 | 100.0 | 99.6 | 100.0 | 100.0 | 100.0 | 100.0 | 98.8 | 81.4 | 100.0 | 99.0 | 98.3 | 98.7 | 100.0 | 99.0 |
| UWaveGestureLibrary | 41.2 | 87.8 | 85.9 | 85.6 | 85.6 | 83.4 | 85.9 | 87.5 | 85.3 | 85.0 | 86.6 | 82.1 | 80.3 | 85.3 | 85.8 | 86.7 | 86.7 | 90.0 |
| Average Accuracy | 48.6 | 71.8 | 70.9 | 71.5 | 72.1 | 70.8 | 71.1 | 72.7 | 70.7 | 71.0 | 73.0 | 67.5 | 70.4 | 73.6 | 72.5 | 74.2 | 75.3 | 74.4 |

# G   GenAI Usage Disclosure

The authors utilized ChatGPT for grammar checking and language polishing of the manuscript. No content was generated by generative AI tools.

Table 13: Full results of the anomaly detection task. Metrics include Precision (P), Recall (R), and F1-score.

| | SMD | | | MSL | | | SMAP | | | SWaT | | | PSM | | | Avg F1 |
|---|---|---|---|---|---|---|---|---|---|---|---|---|---|---|---|---|
| | P | R | F1 | P | R | F1 | P | R | F1 | P | R | F1 | P | R | F1 | |
| LSTM | 78.52 | 65.47 | 71.41 | 78.04 | 86.22 | 81.93 | 91.06 | 57.49 | 70.48 | 78.06 | 91.72 | 84.34 | 69.24 | 99.53 | 81.67 | 77.97 |
| Transformer | 83.58 | 76.13 | 79.56 | 71.57 | 87.37 | 78.68 | 89.37 | 57.12 | 69.70 | 68.84 | 96.53 | 80.37 | 62.75 | 96.56 | 76.07 | 76.88 |
| LogTrans | 83.46 | 70.13 | 76.21 | 73.05 | 87.37 | 79.57 | 89.15 | 57.59 | 69.97 | 68.67 | 97.32 | 80.52 | 63.06 | 98.00 | 76.74 | 76.60 |
| TCN | 84.06 | 79.07 | 81.49 | 75.11 | 82.44 | 78.60 | 86.90 | 59.23 | 70.45 | 76.59 | 95.71 | 85.09 | 54.59 | 99.77 | 70.57 | 77.24 |
| Reformer | 82.58 | 69.24 | 75.32 | 85.51 | 83.31 | 84.40 | 90.91 | 57.44 | 70.40 | 72.50 | 96.53 | 82.80 | 59.93 | 95.38 | 73.61 | 77.31 |
| Informer | 86.60 | 77.23 | 81.65 | 81.77 | 86.48 | 84.06 | 90.11 | 57.13 | 69.92 | 70.29 | 96.75 | 81.43 | 64.27 | 96.33 | 77.10 | 78.83 |
| Anomaly* | 88.91 | 82.23 | 85.49 | 79.61 | 87.37 | 83.31 | 91.85 | 58.11 | 71.18 | 72.51 | 97.32 | 83.10 | 68.35 | 94.72 | 79.40 | 80.50 |
| Pyraformer | 85.61 | 80.61 | 83.04 | 83.81 | 85.93 | 84.86 | 92.54 | 57.71 | 71.09 | 87.92 | 96.00 | 91.78 | 71.67 | 96.02 | 82.08 | 82.57 |
| Autoformer | 88.06 | 82.35 | 85.11 | 77.27 | 80.92 | 79.05 | 90.40 | 58.62 | 71.12 | 89.85 | 95.81 | 92.74 | 99.08 | 88.15 | 93.29 | 84.26 |
| LSSL | 78.51 | 65.32 | 71.31 | 77.55 | 88.18 | 82.53 | 89.43 | 53.43 | 66.90 | 79.05 | 93.72 | 85.76 | 66.02 | 92.93 | 77.20 | 76.74 |
| Stationary | 88.33 | 81.21 | 84.62 | 68.55 | 89.14 | 77.50 | 89.37 | 59.02 | 71.09 | 68.03 | 96.75 | 79.88 | 97.82 | 96.76 | 97.29 | 82.08 |
| DLinear | 83.62 | 71.52 | 77.10 | 84.34 | 85.42 | 84.88 | 92.32 | 55.41 | 69.26 | 80.91 | 95.30 | 87.52 | 98.28 | 89.26 | 93.55 | 82.46 |
| ETSformer | 87.44 | 79.23 | 83.13 | 85.13 | 84.93 | 85.03 | 92.25 | 55.75 | 69.50 | 90.02 | 80.36 | 84.91 | 99.31 | 85.28 | 91.76 | 82.87 |
| LightTS | 87.10 | 78.42 | 82.53 | 82.40 | 75.78 | 78.95 | 92.58 | 55.27 | 69.21 | 91.98 | 94.72 | 93.33 | 98.37 | 95.97 | 97.15 | 84.23 |
| FEDformer | 87.95 | 82.39 | 85.08 | 77.14 | 80.07 | 78.57 | 90.47 | 58.10 | 70.76 | 90.17 | 96.42 | 93.19 | 97.31 | 97.16 | 97.23 | 84.97 |
| TimesNet (I) | 87.76 | 82.63 | 85.12 | 82.97 | 85.42 | 84.18 | 91.50 | 57.80 | 70.85 | 88.31 | 96.24 | 92.10 | 98.22 | 92.21 | 95.21 | 85.49 |
| TimesNet (R) | 88.66 | 83.14 | 85.81 | 83.92 | 86.42 | 85.15 | 92.52 | 58.29 | 71.52 | 86.76 | 97.32 | 91.74 | 98.19 | 96.76 | 97.47 | 86.34 |
| CrossFormer | 83.60 | 76.61 | 79.70 | 84.68 | 83.71 | 84.19 | 92.04 | 55.37 | 69.14 | 88.49 | 93.48 | 90.92 | 97.16 | 89.73 | 93.30 | 83.45 |
| PatchTST | 87.42 | 81.65 | 84.44 | 84.07 | 86.23 | 85.14 | 92.43 | 57.51 | 70.91 | 80.70 | 94.93 | 87.24 | 98.87 | 93.99 | 96.37 | 84.82 |
| ModernTCN | 87.86 | 83.85 | 85.81 | 83.94 | 85.93 | 84.92 | 93.17 | 57.69 | 71.26 | 91.83 | 95.98 | 93.86 | 98.09 | 96.38 | 97.23 | 86.62 |
| Chimera | 87.74 | 83.29 | 85.46 | 84.01 | 86.83 | 85.39 | 93.05 | 58.12 | 71.55 | 92.18 | 95.93 | 94.01 | 97.30 | 96.19 | 96.74 | 86.69 |
| Ours | 86.31 | 81.33 | 83.74 | 85.66 | 84.31 | 84.97 | 92.24 | 64.86 | 76.17 | 91.76 | 95.29 | 93.94 | 98.20 | 96.67 | 97.42 | 87.24 |

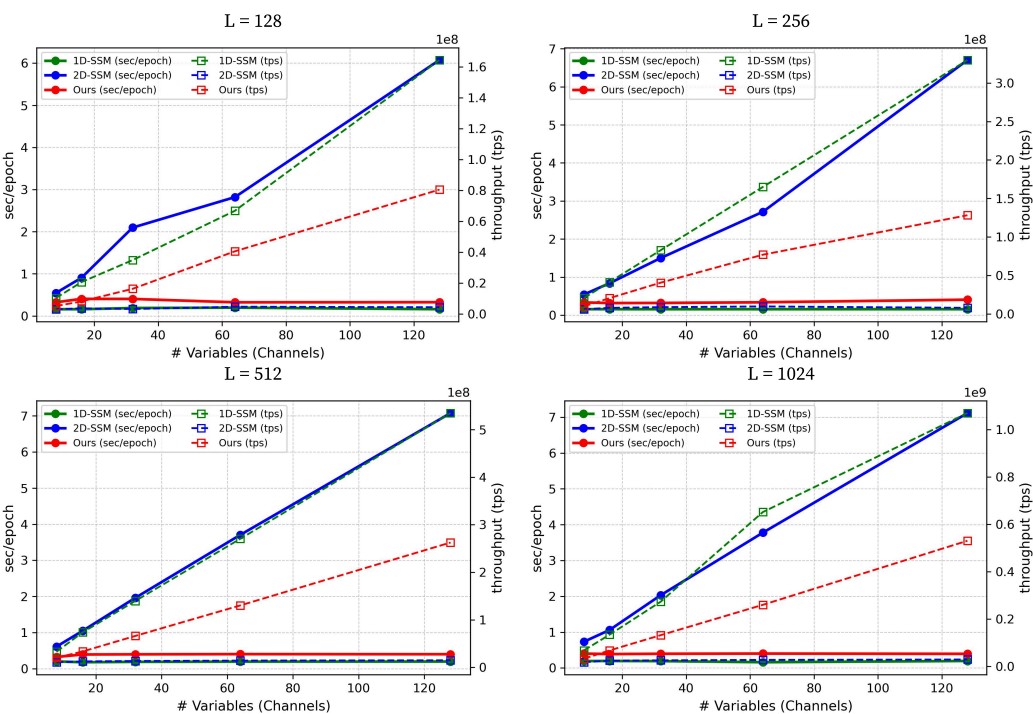

Figure 5: Efficiency analysis with respect to the number of variables under various sequence lengths.

