# OpenReview forum: "Mamba Unchained: A Permutation-Invariant Approach to Multivariate Time Series"
_ICLR.cc/2026/Conference — Submitted to ICLR 2026_

### Official Review · Reviewer_dD7t · 2025-10-17

**Soundness:** 2
**Presentation:** 3
**Contribution:** 2
**Rating:** 4
**Confidence:** 4

**Summary:**

This paper proposes Mamba Unchained, a variable-invariant two-dimensional state space model (2D SSM) designed for multivariate time series analysis. Unlike conventional 1D or 2D SSMs that depend on variable ordering or sequential variable-axis scans, the proposed method introduces a global permutation-invariant pooling mechanism to model cross-variable dependencies. This design allows simultaneous and order-free variable-axis updates, improving both scalability and robustness. The model further adopts a multi-branch architecture incorporating long-term, short-term, and frequency-domain pathways, fused via an adaptive gating mechanism. Experiments across forecasting, classification, and anomaly detection tasks show model’s good performance.

**Strengths:**

1.The proposed variable-invariant 2D SSM fills a key gap in existing literature by resolving the artificial variable ordering dependence of conventional 2D SSMs (e.g., Chimera)；

2.The multi-branch framework (long/short-horizon temporal + frequency-domain pathways) is well-motivated;

3.The experiments cover diverse tasks (forecasting, classification, anomaly detection) and benchmarks.

**Weaknesses:**

1.The paper lacks an overall architectural diagram that describes each component of the model, as well as its inputs and outputs. Such a diagram is essential for readers to intuitively understand how the variable-invariant 2D SSM, multi-branch pathways (long-horizon temporal, short-horizon temporal, frequency-domain), and lightweight gating mechanism interact with one another, and how raw multivariate time series data flows through these modules to generate final predictions or representations;

2.It is unclear where the results of these baseline models are derived from—whether they are quoted from other existing literatures or obtained through the authors’ own experimental testing;

3.Temporal Convolutional Network (TCN) is a relatively old model architecture, yet the paper shows it achieves surprisingly strong performance—even outperforming more recent advanced models such as iTransformer and PatchTST. This unexpected result requires further explanation: are there specific modifications to the TCN (e.g., improved normalization, adjusted kernel size) in the experiments?

4.The proposed model does not outperform baseline models on tasks such as short-term forecasting and time series classification. For short-term forecasting on the M4 dataset, it only achieves the second-best performance (behind Chimera); for time series classification on UEA datasets, its average accuracy (74.4%) is slightly lower than that of Chimera (75.3%);

5.The description of the experimental setup is unclear, such as the length of the historical window and the number of epochs?

6.The paper does not provide code to verify reproducibility. While a GitHub link is mentioned in Section 5.1 (Footnote 1), the code and running script inside are incomplete;

7.Many formulas lack punctuation marks.

**Questions:**

Please see Weaknesses.

---

> ### Author Response · Authors · 2025-11-21
> **Response to the Reviewer (Part 1)**
>
> We appreciate the reviewer’s valuable comments. We address individual concerns below.
>
> ----
>
> **W1. The paper lacks an overall architectural diagram that describes each component of the model...**
>
> **A.**
> We appreciate the reviewer’s suggestion regarding readability. In the revised version, we added a schematic diagram illustrating the full architecture, including (i) the permutation-invariant 2D SSM and (ii) Mamba architecture. This visual explanation clarifies how information flows across modules and how the descriptor conditions the 2D SSM.
>
> ----
>
> **W2. It is unclear where the results of these baseline models are derived from...**
>
> **A.**
> We appreciate the reviewer’s concern regarding the origin of baseline results. We would like to clarify that all reported baseline metrics were directly cited from the original papers, and this is explicitly documented in Section 5. Since these methods provide official benchmark results under their recommended configurations, we followed the standard practice of citing rather than partially re-implementing them, avoiding unfair or mismatched reproduction. To prevent any ambiguity, we will revise Section 5 and the corresponding tables to explicitly tag all cited results with their original sources. Thus, the manuscript already adheres to reproducible reporting standards, and we will improve presentation for added clarity.
>
> ----
>
> **W3. Temporal Convolutional Network (TCN) is a relatively old model architecture...**
>
> **A.**
> We believe there may be a misunderstanding regarding the TCN baseline. The “TCN” reported in our tables is ModernTCN, a recent architecture proposed in ICLR 2024 (spotlight), not the classical TCN from Bai et al. (2018). ModernTCN incorporates channel mixing, global receptive field expansion, and improved normalization strategies, which make it highly competitive against recent transformer and SSM-based models. As stated in Section 5, we directly cite the reported results of ModernTCN under its official benchmark settings.
>
> Because ModernTCN includes these modern enhancements, its strong performance is expected and does not contradict our claims. Rather, it demonstrates that our proposed model outperforms even competitive recent architectures, not outdated CNN-based baselines.

---

> > ### Author Response · Authors · 2025-11-21
> > **Response to the Reviewer (Part 2)**
> >
> > **W4. The proposed model does not outperform baseline models on tasks such as short-term forecasting and time series classification...**
> >
> > **A.**
> > We acknowledge the reviewer’s observation that our method is not the top performer on M4 and achieves slightly lower accuracy than Chimera on UEA benchmarks. This behavior is consistent with our architectural motivation and is explicitly discussed in Section 4 and Section 5.3.
> >
> > Our model is specifically designed to address variable-order dependence and multivariate scaling effects, which yield the greatest benefits when cross-variable temporal interactions drive predictive performance. In contrast:
> > - M4 is univariate $(C=1)$, where permutation invariance offers no advantage, and our model behaves similarly to a 1D SSM baseline.
> > - UEA classification tasks are multivariate but often dominated by instance-level global patterns (shape matching) rather than temporal cross-variable dynamics. Furthermore, many UEA datasets are normalized/preprocessed so that inter-channel correlations play a limited predictive role. Therefore, the benefit of variable-invariant modeling becomes less pronounced for UEA, explaining the small accuracy gap relative to Chimera.
> >
> > Thus, these cases do not contradict the contribution of our work; instead, they confirm that cross-variable invariance is most impactful for multivariate forecasting with strong inter-variable coupling, where our model consistently outperforms baselines (e.g., ECL, ETTh1/2, Traffic, and controlled VAR simulation).
> >
> > To further support where permutation invariance becomes beneficial, we conducted a controlled scaling experiment by varying the number of variables $C$ in a small-world VAR(1) simulation.
> >
> > | Method   | #C  | sec/epoch | MAE   | MAPE  |
> > |----------|-----|-----------|-------|-------|
> > | **Ours** | 16  | 6.2s      | 0.106 | 2.465 |
> > |          | 32  | 6.1s      | 0.098 | 2.146 |
> > |          | 64  | 6.1s      | 0.093 | 2.007 |
> > |          | 128 | 6.4s      | 0.096 | 2.313 |
> > |          | 256 | 6.3s      | 0.097 | 2.663 |
> > | **2D-SSM** | 16  | 13.4s     | 0.105 | 2.603 |
> > |           | 32  | 19.4s     | 0.098 | 2.101 |
> > |           | 64  | 23.4s     | 0.093 | 1.954 |
> > |           | 128 | 51.0s     | 0.096 | 2.489 |
> > |           | 256 | 88.8s     | 0.097 | 2.779 |
> >
> > As shown in the Table, both our model and 2D-SSM reach their best accuracy around $C=64$, indicating that both models effectively capture cross-variable coupling. However, the latency behavior is markedly different: our latency remains nearly constant (≈6.1s/epoch) as $C$ increases from 16 to 256, while 2D-SSM latency grows almost linearly from 13.4s to 88.8s because it must sequentially scan along the variable axis. This demonstrates that the proposed permutation-invariant pooling enables $O(1)$ runtime w.r.t. $C$, whereas existing 2D-SSMs incur $O(C)$ cost even when prediction accuracy is comparable. This directly supports our design motivation and explains why the benefit is less visible on low-$C$ regimes such as M4 and some UEA datasets.
> >
> > ----
> >
> > **W5 & 6. The description of the experimental setup is unclear; The paper does not provide code to verify reproducibility.**
> >
> > **A.**
> > **Regarding experimental setup clarity.** Our training follows the standard settings used in publicly available time-series libraries (e.g., forecasting benchmarks in official SSM/Transformer repositories). Specifically, the historical window was selected from $L\in${96,192,336,720\} following these baselines, and all models were trained for 100 epochs with early stopping. Since all hyperparameters strictly follow these released benchmarks, we did not duplicate the full tables in the manuscript.
> >
> > **Regarding reproducibility.** The model implementation is already fully released. The only missing component is the training script, which is identical in structure to the aforementioned public libraries and will be released in cleaned form. We will upload the same training pipelines used in our experiments to ensure full reproducibility.
> >
> > ----
> >
> > **W7. Many formulas lack punctuation marks.**
> >
> > **A.**
> > Thank you for pointing this out. We have carefully revised the manuscript and added appropriate punctuation to all mathematical expressions and equations to improve readability and consistency with standard notation.

---

### Official Review · Reviewer_uCBS · 2025-10-31

**Soundness:** 3
**Presentation:** 3
**Contribution:** 2
**Rating:** 4
**Confidence:** 5

**Summary:**

This paper proposes a variable-invariant two-dimensional state space model that eliminates dependency on variable ordering by regulating temporal dynamics through the use of globally permutation-invariant descriptors. Extensive experiments demonstrate that the proposed model consistently outperforms state-of-the-art benchmark models in prediction, classification, and anomaly detection tasks.

**Strengths:**

1. The study is rich in experimental content. The paper conducted a substantial number of qualitative and quantitative experiments to validate the effectiveness of the methodology.
2. The paper possesses a robust theoretical foundation, thereby providing theoretical validation of the method's interpretability.

**Weaknesses:**

1. Lack of comparison with relevant prior work. The abstract asserts that existing SSMs or Transformers suffer from high computational costs or fail to capture cross-variable interactions over time. However, the recent work TimePro [1] has addressed this issue. Notably, TimePro is also based on SSM, rendering it highly relevant to this paper. The authors should clarify distinctions from TimePro and include comparative experiments to validate the novelty and efficacy of their approach.

[1] TimePro: Efficient Multivariate Long-term Time Series Forecasting with Variable-and Time-Aware Hyper-state [ICML'25]

2. Limited readability. The author should provide a schematic diagram of the model to enhance module details and improve the paper's readability.

3. Lack of efficiency metrics. The authors have not provided efficiency comparisons with relevant works such as Simba, Chimera, and TimePro within the paper. According to the Method section, the proposed approach incorporates multi-branching and Fourier transforms, which to some extent reduce its efficiency. The authors should furnish efficiency metrics including comparisons of parameters, FLOPs, and latency.

4. It is recommended that the optimal values in Tables 7 and 8 be bolded to facilitate readers' observation of the specific performance of the proposed method.

**Questions:**

Please refer to the weakness. I will raise my score if the author resolves my issues.

---

> ### Author Response · Authors · 2025-11-21
> **Response to the Reviewer (Part 1)**
>
> We appreciate the reviewer’s valuable comments. We address individual concerns below.
>
> ----
>
> **W1. Lack of comparison with relevant prior work...**
>
> **A.**
> We appreciate the reviewer’s comment pointing out the relevance of TimePro. Both methods address interactions between time and variables, but they do so with different modeling principles and design objectives.
>
> TimePro is designed specifically for long-term forecasting. It augments a 1D SSM with variable- and time-aware hyper-states, which are updated for each variable at every time step to modulate the SSM parameters. This produces expressive variable-specific dynamics that benefit long-horizon prediction, but it maintains variable-indexed state evolution and therefore remains sensitive to variable identity and ordering, lacking permutation invariance.
>
> In contrast, our work focuses on a general formulation for multivariate sequence modeling, rather than a long-horizon forecasting specialization. We start from a 2D SSM perspective, but remove the variable-axis scan entirely, replacing it with a global permutation-invariant descriptor which conditions temporal updates for all variables. This enables permutation invariance, full variable-axis parallelism, and applicability beyond forecasting, including classification and anomaly detection.
>
> To respond to the reviewer’s request, we directly compared both approaches using their official implementations under identical training and inference settings. The forecasting results across six datasets are shown below:
>
> | Dataset   | Ours (MSE) | TimePro (MSE) | Ours (MAE) | TimePro (MAE) |
> |-----------|------------|---------------|------------|---------------|
> | ETTm1     | 0.345      | 0.391         | 0.378      | 0.400         |
> | ETTm2     | 0.258      | 0.281         | 0.317      | 0.326         |
> | ETTh1     | 0.397      | 0.438         | 0.419      | 0.438         |
> | ETTh2     | 0.313      | 0.377         | 0.372      | 0.403         |
> | ECL       | 0.158      | 0.169         | 0.254      | 0.262         |
> | Exchange  | 0.311      | 0.352         | 0.375      | 0.399         |
> | Weather   | 0.227      | 0.251         | 0.264      | 0.276         |
>
> As shown in the experimental tables, our method consistently achieves lower forecasting error than TimePro across all benchmark settings, demonstrating superior predictive performance.
>
> Additionally, we measured computational efficiency on ECL $(H=720, L=96, \text{batch}=16)$ using official implementations. The results show a complementary trend: our model exhibits lower FLOPs because it avoids variable-specific evolution, while TimePro exhibits lower memory due to its hyper-state compression.
>
> | Method   | Memory (MB) | FLOPs (G) |
> |----------|-------------|-----------|
> | Ours | 546.62      | 11.99     |
> | TimePro  | 164.35      | 35.04     |
>
> These results confirm that the two approaches are not substitutes but reflect different modeling strategies: TimePro prioritizes memory-efficient long-horizon forecasting via hyper-state compression, whereas our variable-invariant 2D SSM provides a general-purpose formulation with lower directional computation.
>
> We have clarified this distinction in Section 2 and added the above comparison in the experimental section.
>
> ----
>
> **W2. Limited readability.**
>
> **A.**
> We appreciate the reviewer’s suggestion regarding readability. In the revised version, we added a schematic diagram illustrating the full architecture, including (i) the permutation-invariant 2D SSM and (ii) Mamba architecture. This visual explanation clarifies how information flows across modules and how the descriptor conditions the 2D SSM.

---

> > ### Author Response · Authors · 2025-11-21
> > **Response to the Reviewer (Part 2)**
> >
> > **W3. Lack of efficiency metrics.**
> >
> > **A.**
> > We appreciate the reviewer’s request for a clearer efficiency comparison. In addition to the results presented in the revised version, we emphasize that our architecture remains efficient despite the use of multi-branch temporal modeling and Fourier transforms.
> >
> > First, the frequency branch introduces no sequential dependency: its FFT cost is $O(L\log L)$, which runs in parallel and is dominated by the temporal SSM scan $O(L⋅C⋅d)$. While FFT requires an additional activation buffer, this overhead is bounded by the hidden dimension and window length and does not scale with the number of variables $C$. Thus, although the frequency branch slightly increases peak memory, it does so without affecting latency or scan complexity, and remains negligible relative to the removed variable-axis recurrence.
> >
> > Second, we provide quantitative measurements of FLOPs, memory, and latency using public official implementations, following reproducibility principles. Since Chimera does not release an executable implementation, its exact FLOPs and memory cannot be measured reliably; we therefore report results only for models with official implementations (e.g., TimePro, S-Mamba, iTransformer, PatchTST). Measurements on ECL ($L=96$, $H=720$, batch$=16$) are summarized below:
> >
> > | Method       | Memory (MB) | FLOPs (G) |
> > |--------------|-------------|-----------|
> > | Ours     | 546.62      | 11.99     |
> > | TimePro      | 164.35      | 35.04     |
> > | S-Mamba      | 270.76      | 96.13     |
> > | PatchTST     | 675.26      | 85.64     |
> > | iTransformer | 1155.14     | 73.20     |
> >
> > These results show a complementary trend: our method has the lowest FLOPs and fastest latency due to removal of variable-axis state evolution, while TimePro exhibits lower memory usage via hyper-state compression.
> >
> > ----
> >
> > **W4. It is recommended that the optimal values in Tables 7 and 8 be bolded to facilitate readers' observation of the specific performance of the proposed method.**
> >
> > **A.**
> > Thank you for the helpful suggestion. In the revised manuscript, we have bolded all optimal values in Tables 7 and 8 to improve readability and enable easier comparison across methods.

---

### Official Review · Reviewer_rRqE · 2025-10-31

**Soundness:** 2
**Presentation:** 3
**Contribution:** 3
**Rating:** 6
**Confidence:** 4

**Summary:**

The paper proposes a permutation-invariant 2D state space model (2D SSM) for multivariate time series. Instead of sequentially scanning variables, it introduces a global permutation-invariant descriptor ψ(t) to jointly model temporal and variable dynamics, enabling full parallelism across variables. Built on this, a multi-branch Mamba architecture with long-/short-term and frequency-domain branches is designed, fused via lightweight gating. Experiments show superior performance, efficiency, and robustness to variable permutations across forecasting, classification, and anomaly detection tasks.

**Strengths:**

1. The problem is clearly defined, and the design is simple yet effective. By replacing variable-axis recursion with a permutation-invariant global aggregation, the method directly addresses the core challenge of “no natural variable order” in applying 2D SSMs to multivariate time series, while reducing 2D scanning to 1D temporal scanning plus parallel aggregation.

2. The multi-domain modeling design is well-motivated. Long-/short-term temporal branches capture multi-scale dynamics via different discretization steps, while the frequency-domain branch models spectral structures. These are adaptively fused through gating, achieving strong expressiveness with low implementation cost.

**Weaknesses:**

1. Replacing variable-axis scanning with a global aggregation is the core of this work. In the paper, what specific forms of ϕ (e.g., mean pooling, gating, or set-attention) are actually used as the default setting? How do different choices of ϕ affect GPU throughput and peak memory usage? Has a systematic ablation study been conducted?
2. The advantage is less evident in univariate or low-dimensional settings. On the M4 (single-channel) dataset, the method is suboptimal, and the authors acknowledge that the benefit of permutation invariance is limited in this case. It is suggested to more systematically characterize the performance transition as the number of variables C varies.
3. The rationale for choosing the frequency-domain branch hyperparameter Δf is unclear. Table 4 shows that a smaller Δf yields better results, but no adaptive mechanism or theoretical justification is provided, despite the large variation in spectral distributions across datasets.

**Questions:**

1. Does the permutation-invariant conditional independence assumption have any side effects? Are there local interactions among variables, such as pairwise or higher-order relationships, that might be disrupted by the parameter-sharing constraint imposed in the model? What would happen if the parameters were not shared? Have any related experiments been conducted?
2. The paper only compares the efficiency between the proposed global aggregation and the 2D-SSM baseline, without presenting a direct comparison of FLOPs and memory usage against existing methods such as Chimera. Could the authors provide a detailed mathematical derivation of the computational complexity?

---

> ### Author Response · Authors · 2025-11-21
> **Response to the Reviewer (Part 1)**
>
> We appreciate the reviewer’s valuable comments. We address individual concerns below.
>
> ---
>
> **W1. Replacing variable-axis scanning with a global aggregation...**
>
> **A.**
> **Choice of the aggregation function.**
> Our formulation allows any permutation-invariant aggregation function $\psi$, including mean/sum pooling or attention-based pooling. In practice, we select a different default $\psi$ per task based on the inductive bias and computational requirements of each setting.
> - Forecasting (mean pooling): forecasting benchmarks such as ETTh1/ETTm1 require stable, low-variance conditioning and are highly sensitive to error accumulation over many prediction steps. In this case, mean pooling offers the best trade-off between stability and efficiency. That is, variables share strong global seasonal trends, and simple averaging provides a robust global descriptor that suppresses channel-wise noise.
> - Classification/anomaly detection (attention pooling): these tasks rely on discriminative, often sparse cross-variable structure, where only a subset of channels may drive the label or indicate an anomaly. Here, mean pooling tends to dilute such signals, and an attention-based $\psi$ that selectively up-weights informative variables provides a clear benefit.
>
> This task-dependent selection is consistent with prior work in set learning, where pooling architectures are typically chosen to match the complexity and signal structure of the target task.
>
> **GPU throughput and memory usage: theoretical computation.**
> Furthermore, to address the reviewer’s concern about throughput and memory, we provide the following analytical comparison of the operators used in the ablations. Here, $C$ is the number of variables and $d$ is the hidden dimension.
>
> | Aggregation ψ | Work per timestep | Parallel span | Additional peak memory | Notes                                           |
> | ------------- | ----------------- | ------------- | ---------------------- | ----------------------------------------------- |
> | Mean          | O(C·d)            | O(log C)      | O(d)                   | Lightweight; stable; forecasting default        |
> | Sum           | O(C·d)            | O(log C)      | O(d)                   | Similar to mean; no scaling factor              |
> | Attention     | O(C·d²)           | O(log C)      | O(C·d)                 | More expressive; classification/anomaly default |
>
> Key clarifications:
> - Mean and sum pooling introduce only a single accumulator vector and do not require storing intermediate activations: minimal peak memory.
> - Attention-based pooling, however, requires storing per-variable keys/values and attention scores: peak memory grows linearly with $C$.
>
> This theoretical scaling explains why mean pooling is chosen for forecasting: it avoids quadratic projection costs and large intermediate tensors.
>
> **Ablation on the choice of aggregation.**
> We thank the reviewer for raising this point. We have now included a systematic ablation study comparing different permutation-invariant aggregation functions $\psi$.
>
> | $\psi$ (ETTh1) | MSE   | MAE   |
> |-----------------|-------|-------|
> | Mean            | 0.397 | 0.419 |
> | Attention       | 0.409 | 0.426 |
>
> | $\psi$ (ETTm1) | MSE   | MAE   |
> |-----------------|-------|-------|
> | Mean            | 0.345 | 0.378 |
> | Attention       | 0.347 | 0.381 |
>
>
>
>
>
> Mean pooling consistently outperforms attention-based pooling across both datasets. This supports our design choice: forecasting tasks benefit more from stable, low-variance, parameter-free aggregation, with minimal computational overhead. This ablation confirms that the default choice of mean pooling for forecasting tasks is both empirically justified and computationally optimal.
>
> Our updated manuscript includes (i) a clear explanation of the task-dependent selection of $\psi$, (ii) a detailed complexity and memory comparison table, and (iii) ETTh1/ETTm1 ablation results. These additions clarify the behavior of each aggregation operator and demonstrate why our default choices are both principled and empirically justified.

---

> ### Author Response · Authors · 2025-11-21
> **Response to the Reviewer (Part 2)**
>
> **W2. The advantage is less evident in univariate or low-dimensional settings...**
>
> **A.**
> To directly address this point, we conducted an additional controlled $C$-scaling experiment on a synthetic VAR(1) multivariate process defined on a Watts–Strogatz graph (same as Table 3 in the main paper). We compare our variable-invariant 2D SSM (“Ours”) against the standard 2D-SSM baseline while varying $C\in${16,32,64,128,256}. The results are summarized in the table below:
>
> | Method   | #C  | sec/epoch | MAE   | MAPE  |
> |----------|-----|-----------|-------|-------|
> | **Ours** | 16  | 6.2s      | 0.106 | 2.465 |
> |          | 32  | 6.1s      | 0.098 | 2.146 |
> |          | 64  | 6.1s      | 0.093 | 2.007 |
> |          | 128 | 6.4s      | 0.096 | 2.313 |
> |          | 256 | 6.3s      | 0.097 | 2.663 |
> | **2D-SSM** | 16  | 13.4s     | 0.105 | 2.603 |
> |           | 32  | 19.4s     | 0.098 | 2.101 |
> |           | 64  | 23.4s     | 0.093 | 1.954 |
> |           | 128 | 51.0s     | 0.096 | 2.489 |
> |           | 256 | 88.8s     | 0.097 | 2.779 |
>
> Key findings:
> - **Accuracy remains stable across all dimensionalities.** For all $C$, our model matches the MAE/MAPE performance of 2D-SSM, confirming that the use of a permutation-invariant global descriptor does not harm expressiveness, even at high dimensionality.
> - **Our method shows constant-time scaling with respect to C.** Although the global pooling operation has theoretical complexity $O(C)$, it is executed as a single fully-parallel GPU kernel. Consequently, the actual training time remains nearly constant (≈6.1–6.4s per epoch) across all values of $C$, in stark contrast to the 2D-SSM baseline whose runtime grows linearly from 13.4s to 88.8s. This empirically validates that variable-axis updates no longer incur sequential overhead.
> - **The computational advantage grows rapidly with dimensionality.**
> Since 2D-SSM requires a variable-axis recursive pass, its training time increases almost linearly with $C$. In contrast, our model maintains stable runtime due to parallel pooling. At $C=256$, our method is approximately 14× faster while achieving the same accuracy. This confirms that the benefit of the proposed variable-invariant formulation is strictly amplified in high-dimensional multivariate settings.
>
> These results provide direct empirical support that our variable-invariant formulation yields increasing advantage as the number of variables grows, both in computational efficiency and robustness to variable dimensionality. We will include this analysis, together with the above table and accompanying discussion, in the final version.
>
> ----
> **W3. The rationale for choosing the frequency-domain branch hyperparameter $\Delta_f$ is unclear....**
>
> **A.**
> We clarify that $\Delta_f$ is not a frequency subsampling stride, but instead the discretization step size of the continuous-time SSM applied in the frequency domain. The semantics of the state evolution differ from the temporal SSM: states evolve across frequency bands rather than across time, and the update behaves as a spectral ODE discretized with step $\Delta_f$.
>
> Real-world time-series signals exhibit low-frequency energy dominance, while high-frequency components contain less power yet often encode critical oscillatory or transient information. Using a large $\Delta_f$ results in a coarse and unstable discretization:
> - The exponential term overly attenuates high-frequency modes, numerical aliasing increases, and eigenmode scaling becomes inaccurate.
>
> Setting $\Delta_f$ small effectively stabilizes the SSM update, suppresses aliasing, and increases the effective resolution in high-frequency ranges. This behavior aligns with both (i) classical continuous-time SSM discretization analysis and (ii) the known spectral imbalance of real-world datasets.
>
> Our ablation in Table 5 shows that $\Delta_f$ in the range [0.001–0.01] achieves the best balance between numerical stability and spectral fidelity across ETTh1 and ETTm1. Because $\Delta_f$ governs the internal numerical dynamics of the frequency-domain SSM rather than dataset-specific spectral properties, this optimal range remains stable across datasets.
>
> We will clarify the role of $\Delta_f$ as a discretization step size, explain why a small $\Delta_f$ is beneficial in spectral SSMs.

---

> ### Author Response · Authors · 2025-11-21
> **Response to the Reviewer (Part 3)**
>
> **Q1. Does the permutation-invariant conditional independence assumption have any side effects? ...**
>
> **A.**
> We clarify that our architecture does not assume that variables become independent once conditioned on the global descriptor. Instead, the proposed dynamics explicitly combine local variable-specific states and global cross-variable context, as seen in the vertical update equation:
> \begin{align}
> \frac{\partial h_v (t, c)}{\partial t} =A_vh_v(t, c) + A_{v\psi}\psi(t) + A_{vh}h_h(t, c) + B_vx(t,c),
> \end{align}
> where $h_v(t,c)$ and $h_h(t,c)$ are variable-specific vertical and horizontal states, and $\psi$ is a nonlinear permutation-invariant function of all variables. Thus, each variable update is influenced by:
> - Local vertical dynamics $A_vh_v(t, c)$;
> - Local temporal state $A_{vh}h_h(t, c)$, which already aggregates longer-range cross variable effects;
> - Global interactions encoded in $\psi$;
> - Local input $x(t, c)$.
>
> As a result, cross-variable influences are not removed, but rather mediated through $\psi$ and $h_h(t, c)$. This structure enables rich pairwise and higher-order dependencies to be captured in the shared latent pathway, while maintaining permutation invariance.
>
> To further verify that parameter sharing does not harm local interaction modeling, we highlight the controlled VAR(1) small-world simulation (Table 3). This setting contains explicit structured pairwise dependencies between variables. Importantly, the 2D-SSM baseline—which performs an order-dependent sequential variable scan and therefore does not share parameters across variables—achieves the same MAE as our parameter-shared model.
>
> This demonstrates empirically that:
> 1. Parameter sharing does not degrade the ability to recover structured pairwise interactions.
> 2. Our model remains fully permutation-invariant, whereas 2D-SSM degrades under variable permutations.
> 3. The proposed mechanism preserves local and global interactions while eliminating variable-order sensitivity.
>
> Together, the equation-level analysis and controlled simulation results confirm that our permutation-invariant formulation does not discard local dependencies; instead, it preserves them through variable-specific states while providing a coherent, order-free, and more efficient mechanism for modeling cross-variable structure.
>
> ----
>
> **Q2. The paper only compares the efficiency between the proposed global aggregation and the 2D-SSM baseline...**
>
> **A.**
> We appreciate the reviewers’ suggestion to provide a quantitative analysis of computational efficiency. To ensure fairness, we benchmarked several publicly available implementations that represent transformer-based and SSM-based multivariate time-series models. All models were evaluated under identical settings: Dataset: ECL; Window length $L = 96$; prediction length $H = 720$; Batch size 16. Since Chimera does not provide an official public implementation, and its published version contains incomplete pseudo-code that prevents reproducibility of exact kernel behavior and caching strategy, we report results only for models whose officially released implementations enable precise FLOPs and memory measurement.
>
> The measured peak GPU memory usage and forward FLOPs are:
> | Method       | Memory (MB) | FLOPs (G) |
> |--------------|-------------|-----------|
> | **Ours**     | 546.62      | 11.99     |
> | TimePro      | 164.35      | 35.04     |
> | S-Mamba      | 270.76      | 96.13     |
> | PatchTST     | 675.26      | 85.64     |
> | iTransformer | 1155.14     | 73.20     |
>
> **Why our method achieves low FLOPs.**
> A classical 2D-SSM updates states along both axes. Let $L$ be sequence length, $C$ the variable dimension, and $d$ the hidden dimension. The 2D scan consists of:
> - Temporal scan: $O(L·C·d)$; Variable-axis recursion: $O(L·C·d)$; Total cost: $O(2L·C·d)$.
>
> However, the second term is inherently sequential across $C$, preventing parallel execution.
>
> **Our formulation removes the sequential variable recursion.**
> We replace the variable-axis recursion with a single permutation-invariant pooled descriptor:
> - Temporal SSM scan: $O(L·C·d)$; Global descriptor: $O(C·d)$; Total cost: $O(L·C·d) + O(C·d)$.
>
> The key distinction is that $O(C·d)$ is fully parallelizable across variables, reducing total execution time to a single SSM temporal scan, without forcing a variable ordering. This design explains why our measured FLOPs are lower than existing SSMs (TimePro, S-Mamba): they maintain time–variable state evolution, whereas our method performs only one directional scan plus a parallel pooling step.

---

### Author Response · Authors · 2025-11-26
**General Response**

We sincerely appreciate the reviewers’ careful reading of our manuscript and their insightful comments. The feedback has significantly contributed to improving both the clarity and completeness of our work. In the revised version, we addressed all major concerns through (i) additional theoretical explanations, (ii) extended experiments, and (iii) improved presentation and readability. The key updates are summarized as follows:

**1. Clarified the role of permutation-invariant aggregation and its task-dependent choice.**

We provided a comprehensive explanation of why different aggregation functions (mean/attention) are suitable for different tasks. A new ablation study (Table 5. Case III) demonstrates that mean pooling is optimal for forecasting tasks due to its stability and low variance. We additionally included a theoretical comparison of computational and memory costs across aggregation operators, showing that attention-based pooling incurs higher peak memory proportional to the number of variables. This clarification is now reflected in both the revised paper (Section 6) and Appendix E.

**2. Added controlled experiments demonstrating the scalability advantage of our formulation.**

To evaluate how model benefits vary with dimensionality, we conducted a new VAR(1) controlled simulation by varying the number of variables $C\in${16,32,64,128,256} (Table 4). The results show that our method preserves accuracy while the training time remains nearly constant, whereas the 2D-SSM baseline grows linearly with $C$. This provides direct empirical evidence that the proposed variable-invariant update leads to increasing advantage in high-dimensional multivariate settings.

**3. Clarified the interpretation and tuning of the frequency-domain step size $\Delta_f$.**

We revised the text to explain that $\Delta_f$ is not a subsampling stride but a discretization step of a continuous-time frequency-domain SSM (Section 4.2.2). A small $\Delta_f$ ensures stable propagation of high-frequency modes, preventing aliasing and excessive attenuation. Our ablation shows that $\Delta_f\in$[0.001, 0.01] yields stable behavior across datasets. This explanation resolves ambiguity between temporal and spectral discretization semantics.

**4. Expanded efficiency comparison to modern SSM/Transformer baselines.**

We benchmarked several officially released implementations (TimePro, S-Mamba, PatchTST, iTransformer) and reported FLOPs and memory under identical experimental settings (Section 6 and Appendix E.3). The empirical analysis supports that our model attains the lowest FLOPs due to the removal of variable-axis scanning, with complementary memory behavior relative to compression-based models such as TimePro.

**5. Improved readability and reproducibility.**
- A new schematic diagram illustrating the overall architecture has been added (Figure 2).
- Formula punctuation and mathematical formatting have been corrected.
- All tables now bold optimal values for clarity.
- The baseline values are explicitly cited in Section 5, and the training script will be uploaded in cleaned form.

Collectively, these revisions strengthen the theoretical grounding, empirical support, and readability of our work. We thank the reviewers again for their constructive feedback and believe that the revised manuscript more clearly demonstrates the novelty, practicality, and scalability of the proposed variable-invariant 2D SSM framework.

---

### Meta-Review · Area_Chair_4vCN · 2026-01-07

**Summary:**

This paper proposes a variable-invariant two-dimensional state space model that eliminates variable ordering dependence by leveraging a global, permutation-invariant descriptor to condition temporal dynamics. I have carefully read the revised manuscript, the reviews, and the rebuttal. I appreciate the authors’ efforts in addressing several reviewer concerns, particularly by adding additional ablations, efficiency analyses, and scalability experiments. However, after considering the full discussion, I do not believe the paper meets the acceptance bar for ICLR in its current form.

The primary reason for rejection is that the performance improvement over the most relevant baseline, Chimera, remains very limited, based on the reported results in Table 2 and 10. In comparison, the proposed permutation-invariant design does not demonstrate clear or consistent advantages in terms of forecasting accuracy across key benchmarks. In addition, the experimental setups are somehow not clearly presented, which further weaken the empirical evidence.

Overall, although the topic is significant and the idea is reasonable, the limited improvement over Chimera prevents the work from making a sufficiently strong recommendation for acceptance at ICLR. I therefore recommend rejection.

**Reviewer Concerns:**

Insufficient experimental evalution issues are still outstanding after the rebuttal. Moreover, the performance improvement is limited compared  to the baselines.

**Reviewer Scores:**

All three reviewers would not have changed their scores (6, 4, 4) even they had been able to participate fully in the discussion.

---

### Decision · Program_Chairs · 2026-01-26

Reject